# Quantum Biotech and Internet of Virus Things: Towards a Theoretical Framework

**Prafulla Kumar Padhi *** and **Feranando Charrua-Santos**

Eletromechanical Department, University of Beira Interior, 6201-001 Covilha, Portugal; bigares@ubi.pt
* Correspondence: pkpadhi20162017@gmail.com or prafulla.k.padhi@ubi.pt

**Abstract:** Quantumization, the process of converting information into quantum (qubit) format, is a key enabler for propelling a new and distinct infrastructure in the pharmaceutical space. Quantum messenger RNA (QmRNA) technology, an indispensable constituent of quantum biotech (QB), is a compelling alternative to conventional vaccine methods because of its capacity for rapid development, high efficacy, and low-cost manufacturing to combat infectious diseases. Internet of Virus Things (IoVT), a biological version of Internet of Things (IoT), comprises applications to fight against pandemics and provides effective vaccine administration. The integration of QB and IoVT constitutes the QBIoVT system to advance the prospect of QmRNA vaccine discovery within a few days. This research disseminates the QBIoVT system paradigm, including architectural aspects, priority areas, challenges, applications, and QmRNA research engine design to accelerate QmRNA vaccines discovery. A comprehensive review of the literature was accomplished, and a context-centered methodology was applied to the QBIoVT paradigm forensic investigations to impel QmRNA vaccine discovery. Based on the above rumination, the principal motive for this study was to develop a novel QBIoVT theoretical framework which has not been produced through earlier theories. The proposed framework shall inspire future QBIoVT system research activities to improve pandemics detection and protection.

**Keywords:** mRNA technology; quantum information system; quantum computing; quantum intelligence; QBIoVT system paradigm; quantumization; QmRNA vaccine discovery





## 1. Introduction

An imaginative and prescient quantum theory (QT), also referred to as quantum mechanics (QM), is part of quantum field theory (QFT) [1] (p. 1877) that expounds the physical properties of nature on an atomic scope and explicates quantum physics as the golden toddler of modern science. An untenable and flummoxing aspect of QM is the wave–particle duality phenomenon wherein objects function like waves and particles. One of the supreme scientific triumphs of the 20th century was the development of QT and the subsequent development of the primary wave of applications, for example, computers, smartphones, transistors, superconductors, laser, and many others. Quantum Information Systems (QIS) [2] represent three foremost areas: quantum computation, quantum control, and quantum communication, and these disciplines are associated with each other based on the concepts of QM. Quantum computing (QC), quantum sensors, and quantum simulation are examples of quantum technologies [3], where properties of QM are important. Artificial intelligence (AI), quantum computing (QC), and machine learning (ML) are three technologies that have unimaginable potential. Therefore, they have integrated together to attain a quantum benefit, in which complex algorithms may be computed inexorably quicker than with the classical supercomputer, giving upward thrust to a game-changer concept called quantum machine learning (QML) [4] (pp. 1–2). Quantum Intelligence (QI—quantum AI) [5] (p. 4), is an emerging ideal solution to improve the notoriously vexing problem and inefficient process of vaccine discovery. One can envision the future role of QI in vaccine

discovery and development, solving perplexing problems with advanced veracity, which will produce a quantum benefit in the introduction of new vaccine discovery.

Pharma businesses have a focal point on messenger RNA (mRNA) [6], a type of RNA found in cells and molecules that carries the genetic information needed to make proteins. A few pharma companies are broadening their horizons with the aid of QI, which is permitting key breakthroughs in data analytic and predictive modeling to boost up knowledge acquisition, providing essential insights into mRNA research and development. mRNA exemplifies a new class of therapeutics for dexterous and adoptable medical applications. The COVID-19 vaccines use nucleic acids called mRNA and do not pose such contamination challenges. The Biotech firm's potential to build on the promise of mRNA technology is a monumental treatment achievement for infectious diseases like COVID-19. In a new era of vaccinology, mRNA vaccines are a significant achievement.

Quantum Biotech (QB) [6] (pp. 1–8) [7] is an emerging essential enabler that these days attracts an in pouring of assidutiy, and its mission is to create a new class of QmRNA vaccine in a few days. QmRNA [7,8] (p. 1) is a novel concept that involves a quantum approach to treating disease with just mRNA and has the plausibility to revolutionize how vaccines are discovered, developed, and manufactured—at a quantum speed, quantum breadth, and quantum scale.

Viruses and bacteria are each infinitesimal living things that may cause disease in humans. While these microbes have some common characteristics, they are additionally very disparate. Researchers agree bacteria and viruses have an immaculate means of communication; in-built engines, sensors, and processing architecture; and efficient information storage capacity. Now bacteria and viruses can join the network. Amelioration of QB, QI, nanotechnology, and virus network are receiving growing assidutiy in scientific research inspiring to constitute the Internet of bio-nano things (IoBNT) [3–6,9,10], [11] (pp. 1–2) encircling communication architecture that embraces biological entities, nanoscale devices, and sensors. The united influence and dynamism of IoT technology and data are adapted to various industrial processes and manufacturing to automate, predict, streamline the process, and improve productivity.

IoT is a network of devices connected and provides intelligence to the IoT community. The Internet of Everything (IoE) is a super set of IoT, constructed on current IoT infrastructure. Like IoBNT, a biological version of the Internet using virus, Internet of Virus Things (IoVT) [9], [10] (pp. 22–23), [11] can endure. Therefore, IoVT is a sub-classification of IoT. In this study, IoT and IoVT terms are used interchangeably. The integration of mRNA technology, QC, QI, and IoVT can subsist to form a QBIoVT system that can advance the discovery of new class QmRNA vaccines in a few days.

The aim of this study was to consolidate the relevant literature and context-centered methodology to develop a novel QBIoVT theoretical framework.

## 1.1. Background and Study Justification

Theory advent and development is a meaningful "Knowledge Nugget" and an array of the scholarly compass. Scholars place an esteemed value on theoretical frameworks. Even though there has been traction in comprehending theory within the IS and QIS domain in contemporary years, the process of theory development is addressed seldom with contributions dawning from other disciplines and inadequate endeavor to meaningfully harmonize them.

Recently, there was extensive research on the discipline of IoT but, so far, not many theories proposed and evolved in the IoT domain. There is prevalent concern that the QIS or IS discipline does not bestow applicable theory development. Theory on QBIoVT system (disciplines that address quantum biotech, mainly associated with QmRNA, quantum operating system (QOS) software solutions, and subjects, specifically philosophy, anthropology, socioeconomic, psychology, and ethical dimensions) domain does not exist within the literature. To date, the QIS discipline is not well understood by academia in other

fields. Furthermore, the development of new theories and the implementation of prevailing theories had been left out within the IS and QIS fields [12] (pp. 2–4, 20).

Currently, the significance of the QBIoVT system topic is gaining traction in the QmRNA vaccine discovery because of the COVID-19 pandemic crisis. A compelling challenge in this regard is how the pharma industry can adapt its traditional technology platform and execute the QBIoVT system innovation to discover a brand-new class of quantum vaccines. To fill the void in the literature, this study bequeaths a nuanced comprehension of holistic steerage to develop a novel QBIoVT theoretical framework

### 1.2. Significant Contributions

This study draws compelling factors for the QBIoVT paradigm, focusing on a theoretical framework that is still unexplored and uncharted.

Pharma research, especially drug/vaccine discovery, is fraught with expensive failures. By predicting faster and better pathways to successful vaccine discovery, averting many of the dead ends is feasible. Integrating QI and cutting-edge biological tools at scale can explicitly outperform instinct and truly enable predictive models. One of the significant insights into recognizing the quantumization, a term coined in this study, research engine method empowers mechanisms to protect against pandemics to build a better society. In pursuing so, it bestows both managerial and theoretical contributions. The compelling contribution embraces the integration of QI and mRNA to form a QmRNA technology platform for QmRNA vaccine discovery.

One of the contributions of the proposed framework is refined know-how of the strategic decision of IoVT via adopting a topological analysis. Evaluation of IoT strategies is from technology push and marketplace pull and manager's strategic intent. This research bestows significant managerial importance for enterprises embracing the IoVT strategy to attain sustainable competitive advantage.

Thus, the significant contribution of this study is the introduction of a QBIoVT system novel theoretical framework that does not prevail, so far, in the literature.

### 1.3. Organization

The rest of this research paper is organized in the following manner: In Section 2, an exhaustive literature review and synthesis is presented. In Section 3, the theoretical foundation outlines the QBIoVT boundaries, IoVT colonnade, relevant theories that support the theoretical framework. A context-centered methodology found to be the best suited for this study is described in Section 4. QBIoVT paradigm is presented outlining QB, specifically QmRNA technologies platform, and IoVT overview, basic architecture as well as architectural components in Section 5. A list of QBIoVT priority areas, challenges, and applications is described in Section 6. In Section 7, a novel QBIoVT theoretical framework and contributions of the framework are discussed. Consolidated lessons learned and future research agenda are explained in Section 8. In Section 9, the theoretical framework was developed which has not been covered by prior theories.

## 2. Literature Review and Synthesis

An exhaustive review of prior, relevant literature is imperative for any academic research topic. A compelling evaluation generates a solid foundation for propelling new know-how to the literature. Furthermore, it facilitates theory development, topics where a plethora of research exists, and uncovers disciplines wherein further research is needed. The major challenge with the QBIoVT system for QmRNA vaccine discovery is the dearth of existing research within the development of a novel theoretical framework. However, extensive literature exists on the topics of IoT, mRNA, QC, and AI. The capacious literature focus on the following several associated topics: (i) QIS/IS disciplines related to developing theoretical framework on QBIoVT system; (ii) QB building blocks specifically on mRNA technology platform on QmRNA vaccine discovery; (iii) QI application for mRNA

platform; (iv) IoVT infrastructure and architecture, trust, security, privacy, compatibility, and regulatory issues; and (v) quantum algorithms and protocols.

### 2.1. Selection of Literature Criteria

The exhaustive literature review for this study was done according to the following criteria:

1. Well-established databases (SCOPUS, IEEE Xplore, MDPI, PubMed, arXiv, IOP) and peer-reviewed reputable international journals from the years 1985 to 2021 were consulted. The web search engine (Google Scholar) indexes the full-text scholarly articles.
2. Books from Wiley, Nature, Springer, and books.google.com were the basis of this research from the years 1985 to 2021.
3. Quantum Technology and System Literature-Searched the databases of Google Scholar, PubMed, SCOPUS, and ResearchGate using the keywords "IoT" and COVID-19.
4. Technology Review published by Massachusetts Institute of Technology (MIT).
5. EPPI-Reviewer includes systematic reviews, meta-analyses, "narrative" reviews, and meta-ethnographies containing over a million items—version 4.11.5.2 (16th Nov 2020).
6. International Conferences proceedings.
7. Ph.D. Theses (Published Ones).
8. Market, Industrial, and Scientific White Papers on connected intelligent industrial trends, perspectives, and applications.

### 2.2. Relationships between Related Topics and Insights

A comprehensive literature review helped in the identification of the key dimensions of the following topics related to QBIoVT systems rationalization:

#### 2.2.1. Theory Development and QIS or IS Discipline

QIS or IS are complex multi-dimensional phenomena [12] (p. 449) and signifies the use of various disciplines to study the development of theory based on the diversity of contexts within the relevant technologies of information. From a conceptual stance, there are no sufficient reasons to accept that QIS is qualitatively different from classical information systems (CIS) or simply called IS [13]. So far, no well-accepted common theoretical basis for an information system is established, even though diversity in IS research and practice has attained significance. In addition, there is no core of a theory well-accepted. From the communication perspective, there is only one kind of data, and physically neutral, and the classical models for QIS secure a specific philosophical and conceptual appeal. The IS and QIS disciplines encircle a range of topics including information communication technology (ICT), operational technology (OT), IoVT, QB, QI, mRNA technology, systems analysis, and design, computer networking, information security, database management, and decision support systems [7–10,12,13]. The three components of information systems include software, hardware, and data—all classified in the category of technology. Comprehending how technical products are created and used within enterprises is a central facet of the IS and QIS research discipline. QIS and IS are relatively new disciplines in the context of QBIoVT.

The literature review topics and related references are listed in Tables 1–4 below, and the subsections of the discussion around each topic bestow significant insights to this research.

**Table 1.** Summary of the literature review on IS/ Quantum Information Systems (QIS) and theoretical framework.

| References/Year | Issues and Remarks on QIS/IS/TF Study |
|---|---|
| Falkenberg et al./1995 [12] | Limitations of IS theory and practice: a case for pluralism. IS Concepts. |
| Lombardi et al./2016 [13] | What is quantum information? |
| Aquilani et al./2020 [14] | Innovation/Value Co-Creation/Theoretical Framework. |
| Nord et al./2019 [15] | IoT/Theoretical Framework (TF)/Systematic Review. |
| Hassan et al./2019 [16] | The process of IS theorizing as a discursive practice. |
| Lim et al./2013 [17] | Theories Used in IS Research: Insights from Complex Network Analysis. |
| Weber, R./2012 [18] | Evaluating and Developing Theories in the IS Discipline. |
| Ridley, G./2006 [19] | Characterizing IS in Australia: A theoretical framework. |
| Leidner, D.E.; Kayworth, T.R./2006 [20] | IS Research: Toward a Theory of IT Culture Conflict. |
| Halawi, L.; McCarthy, R./2006 [21] | IS Research: Toward a Theory of IT Culture Conflict. |
| Swanson, E./1994 [22] | IS Innovation among Organizations. |

### 2.2.2. QmRNA

Albeit one distinct gene requires to do its work, it makes a copy of itself, which is called mRNA. For the past three decades, vaccine scholars have been fascinated and discouraged by the potential of mRNA. Currently, researchers are testing vaccine candidates that use mRNA, without using actual bits of the virus, to generate the immune system to create defensive antibodies. mRNA is a fragile molecule, which means it must be coated in a protective, fatty covering to keep it stable, and the refrigeration conditions have to do with how the mRNA was manufactured and stabilized [23].

A fundamental aspect of human physiology and vital to unleashing the immune system is the tiny particles of genetic code that are key to influencing cells to build proteins. mRNA guides the body's protein production in a much more fixate manner. The experimental vaccines have received approval (emergency authorization of use), for the first time, from regulator bodies, such as Food and Drug Administration (FDA, USA) [23] (p. 3) and European Medicines Agency (EMA) [23] (pp. 3–9) that use mRNA technology. Such approval is a windfall for the mRNA technology, a brand-new platform, which was thought to be wishful thinking for a long time. Researchers at companies like Moderna [24] and Pfizer (BioNTech) [25] vaccine used synthetic mRNA that contains information about the COVID-19 signature spike protein and demonstrated that the vaccine based on mRNA technology platform bestows more than 90% efficacy at preventing symptomatic COVID-19 [23–25]—Pfizer/BioNTech and Moderna hence, the distinction between Moderna and Pfizer/BioNTech vaccine being how the vaccines' synthetic mRNA is manufactured and packaged.

Recently, quantum-inspired genetic algorithms (QGAs) were introduced for the prediction of RNA secondary structures, demonstrating superiority over the existing strategies. Researchers have introduced a new QGA named multi-population assisted quantum genetic algorithm (MAQGA) [26], demonstrating the performances of genetic algorithms with substantial enhancement. The holy grail end-to-end in silico vaccine discovery involves evaluating and breaking down the entire chemical structures of the virus and the cure. Quantum computers, if commercially successful as fault-tolerant systems, will enable end-to-end in silico vaccine discovery. By 2030, with the QmRNA technology, the entire process transforms into a quantum simulation that could eliminate 99.9% of false leads in a fraction of the time empowering QmRNA vaccine discovery in a few days [27].

### 2.2.3. IoVT Archetype

Literature available on IoT, IoE, and other associated topics such as the Internet of Medical Things (IoMT), Internet of Nano Things (IoNT), and Internet of Bio-Nano Things (IoBNT) was reviewed thoroughly.

**Table 2.** Summary of the literature review on Quantum messenger RNA (QmRNA) and mRNA.

| References/Year | Issues and Remarks on mRNA Study Related |
|---|---|
| Jackson et al./Moderna/2020 [24] | messenger RNA Technology Platform/Vaccine Report. |
| Pfizer/BioNTech./2020 [25] | messenger RNA Platform/Covid-19 Vaccine Report. |
| Shi et al./2020 [26] | Multi-Population Assisted Quantum Genetic Algorithm (QGA). |
| Szmuk, R./2020 [27] | Quantum Computing will (eventually) help us discover vaccines in days. |
| Pardi et al./2020 [28] | Recent Advances in mRNA Vaccine Technology. |
| Gómez-Aguado et al./2020 [29] | mRNA: State of the Art and Future Perspectives. |
| Virolle et al./2020 [30] | Plasmid transfer by conjugation in gram-negative bacteria. |
| Kowalski et al./2019 [31] | Advances in Technologies for Therapeutic mRNA Delivery. |
| Patel et al./2019 [32] | mRNA Delivery for Tissue Engineering and Regenerative Applications. |
| Haabeth et al./2018 [33] | mRNA vaccination for human T-cell. |
| Hajj, K.A.; Whitehead, K.A./2017 [34] | Tools for translation: materials for mRNA delivery. |
| Steinle et al./2017 [35] | Application of In Vitro Messenger RNA for Cellular Engineering. |
| Weissman, D./2014 [36] | mRNA transcript therapy review. |
| Petch et al./2012 [37] | Specific mRNA Vaccines against Influenza. |

The IoT is not a concept and no more a niche topic. IoT is a network, the true technology-enabled network of all networks transforming the everyday physical objects that will enrich society. The leverage of IoT in in-door environments could prevent highly infectious diseases from spreading rapidly, specifically during pandemics. Even demure industries have embraced IoT technology. It does not mean IoT rollouts are normally smooth sailing; they pose ample challenges in the adoption and execution of IoT projects. The following are some challenges facing the IoT future: (i) cybersecurity is a concern, and IoT offers potential vulnerability, and the IoT devices are open to attack by botnets; (ii) compatibility is an issue since IoT is a complex ecosystem of various technologies and still lacks standards for quality control.

The life-changing promise of IoMT technology is the capacity to gather, analyze, and communicate massive health data effectively. IoMT tools are utilized to minimize the burden on healthcare systems during the COVID-19 pandemic.

One of the emerging areas of research is on bio-nano things in bacteria or viruses. Even though generally, IoT devices research and development continues, there are sundry applications where microscopic, and non-invasive "things" are required. This concept is called IoNT [37] and enables the artificial nature of IoNT devices that can be pernicious where "Nano Things" deployment may create unwanted and undesirable health issues. The functionalities in the biochemical domain, based on biological cells, "Bio-Nano Things", have the potential to endow applications such as environmental control agents, actuator networks, and intra-body sensing devices. Synthetic biology and nanotechnology tools that empower the biologically embedded computing devices design create the IoBNT [38] archetype.

With the conjunction of mRNA, QC, and QI, the vision of this study focuses on viruses as an IoT device. Even though ample research work has been published on the IoT-related concepts, QBIoVT research-based studies non-existent in the literature. Moreover, the theory associated with QBIoVT adoption and execution is non-existent in the literature.

**Table 3.** Summary of the literature review on Internet of Things (IoT) and related topics.

| References/Year | Remarks on IoT and Related Topics |
| --- | --- |
| Wang et al./2020 [38] | IIoT/Industrial Control Systems. |
| Singh et al./2020 [39] | IoT applications to fight against COVID-19. |
| Fouad et al./2020 [40] | A Nano-biosensors model based on IoBNT. |
| Siemens/2020 [41] | IoT/2050—Industrial IoT. |
| HUAWEI/2020 [42] | IoT/Digital solutions. |
| Singh et al./2020 [43] | IoT Based Blockchain for Temperature Monitoring and Counterfeit Pharmaceutical Prevention. |
| Sekaran et al./2020 [44] | 6G/IoT Automation. |
| CISCO/2020 [45] | Internet of Everything (IoE) Economy. |
| Chamola et al./2000 [46] | IoT/AI Review of the COVID-19 Pandemic. |
| Zhang et al./2019 [47] | 6G/Super IoT Aspects. |
| Clazzer et al./2019 [48] | IoT Applications in 6G proposed. |
| Eppi Center/2019 [49] | IoE. |
| Dai et al./2019 [50] | Blockchain/IoT. |
| Nicolescu, Huth, Radanliev, and Roure/2018) [51] | Value of IoT. |
| Delloite Center of Health Solutions/2018 [52] | Internet of Medical Things (IoMT). |
| Rodrigues et al./2018 [53] | Enabling technologies for the Internet of Health Things. |
| Guo, Chen, and Tsai/2017 [54] | Trust computation models in the IoT systems. |
| BITAG/2016 [55] | IoT Security and Privacy Recommendations. |
| Statista Research Department/2020 [56] | Internet of Things Report (2015–2025). |
| AVANET/ABACUS/2020 [57] | Predictive maintenance with IoT. |
| Sicari, Rizzardi, Grieco, and Coen-Porisini/2015 [58] | Security, privacy, and trust in the IoT. |
| Akyildiz et al./2015 [59] | The Internet of Bio-Nano things, |

### 2.2.4. Quantum Computing (QC) and Quantum Intelligence (QI)

QC revolutionary technology is still at its nascent phase where QM converges with information theory to solve certain specialized computational problems much faster, such as computational chemistry. Simulating QM is a unique application for cutting-edge ML tools. The future applications in the quantum simulation (QS) arena will increasingly gain from the processing of quantum data by ML techniques [60].

QI offers the following possible connections between QC and AI: (i) solving a classical problem of AI from the reference of QC and (ii) exploring the possible synergies between QC and AI. The application of quantum algorithms in AI techniques will increase ML abilities enabling the development of the pharma industry, specifically in vaccine discovery. Furthermore, researchers have advanced QC research bringing quantum-classical computing one step closer by demonstrating a breakthrough in optimized quantum algorithms solving the notorious Fermi–Hubbard model using presently available QC hardware offering a pathway to comprehending and developing novel materials [8,60,61].

**Table 4.** Summary of the Literature Review on quantum computing (QC) and quantum Intelligence (QI).

| References/Year | Remarks on QC/QI Study Related |
| --- | --- |
| Mezzacapo, A./2020 [60] | Connections Between QC and ML in Computational Chemistry. |
| Cade et al./2020 [61] | Quantum Algorithms: Breakthrough Towards Quantum Advantage. |
| Rotta et al./2017 [62] | Quantum Information arXiv. |
| Ristè et al./2017 [63] | Demonstration of quantum advantage in ML. |
| Rotta et al./2016 [64] | Maximum density of quantum information in a scalable CMOS. |
| Wichert, A./2013 [65] | Principles of quantum artificial intelligence. |
| Neven, H./2013 [66] | Launching the quantum artificial intelligence lab. |
| Levine, I.N./2013 [67] | Quantum Chemistry. |
| Ying, M./2014 [68] | Quantum computation, quantum theory and AI. |
| Desurvire, E./2009 [69] | Classical and quantum information theory. |
| Yanofsky, N.S.; Mannucci, M.A./2008 [70] | Quantum computing for computer scientists. |
| Shor, P.W./2003 [71] | Why haven't more quantum algorithms been found? |
| Nielsen, M.A.; Chuang, I.L/2000 [72] | Quantum computation and quantum information. |
| Grover, L.K./1996 [73] | A fast quantum mechanical algorithm for database search. |
| Shor, P.W./1994 [74] | Algorithms for prime factorization and on a quantum computer. |
| Deutsch, D.; Jozsa, R./1992 [75] | Rapid solution of problems by quantum computation. |
| Deutsch, D./1985 [76] | QT, the Church–Turing principle and the universal quantum computer. |

### 3. Theoretical Foundation

*3.1. QBIoVT Foundations, Boundaries, and Research Approach*

The basic axiom of biotechnology is to exploit engineered biotic things for the welfare of humanity. The fundamental credo of molecular biology defines genes to be transcribed from DNA to mRNA and each mRNA in turn translated into proteins. QB is a far-reaching discipline in which organisms, biological phenomena, cells, or cellular components are manifested in the development of QmRNA.

Literature on QmRNA, QC, QI, and IoVT priority areas, challenges, applications, and architectural framework strengthened through the literature review for reinforcing research and to help in the development of the theoretical framework. The following steps were pursued to identify relevant literature (i) usage of the key terms QC, QI, IoT, IoBNT, IoVT, mRNA in Google Scholar; (ii) Google and Firefox were also used for industry white papers related to QC, QI, IoT, IoBNT, IoVT, mRNA applications, and security/privacy/risk issues; (iii) exploration of the leading journals such as IEEE, Science Direct (Elsevier), Emerald Insight; (iv) search of academic research databases was pursued to identify relevant theories; (iv) conference proceedings related to the theme of the research paper.

One of the major problems was the lack of existing research on IoVT, QBIoVT, and protocol treatment protocols for the QmRNA vaccine discovery. However, this study was formulated based on the current and the existing research synthesis related to topics associated with the QBIoVT system. QC and QI are emerging to prove to be beneficial to speed up the process of QmRNA vaccine discovery. Several organizations such as IBM, Moderna, and Pfizer (BioNTech) labs have started to adopt the use of QC, QI, mRNA, and IoT to identify potential vaccines. Furthermore, QML, a subset of QI, has proved effective in the process of QmRNA vaccine discovery.

*3.2. QBIoVT Colonnade*

The QBIoVT system colonnade that includes the IoVT archetype, trust, risk, privacy, and security issues, as shown in Figure 1, is discussed in this section

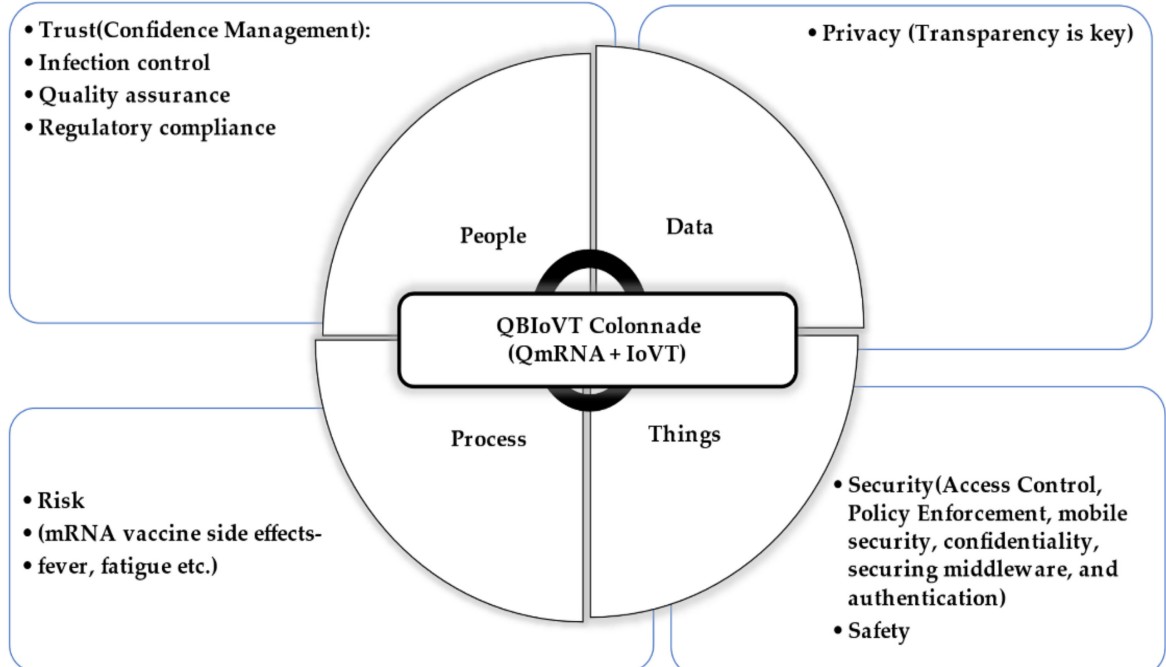

**Figure 1.** Quantum Biotech Internet of Virus Things (QBIoVT)QBIoVT colonnade.

### 3.2.1. IoVT Archetype

Envisage IoVT is conceived as a biological rendition of the IoT. Viruses store and process information as DNA structures called plasmids, that communicate from one organism to the next in a process called conjugation [30] generating a convoluted network mosaic. Viruses consist of a protein shell called a capsid and reproduce as well as survive by seizing control host cell and using its ribosomes to make new viral proteins. Antibiotics do not work against viruses and can be prevented by vaccination. Viruses interact directly with the micro-organisms to bring practical and ethical challenges. These include handling microbes safely and the bio-ethical and biosafety challenges. Privacy issues and ethical considerations related to user data would apply to virus driven IoT systems. As viruses can evolve and behave autonomously, they could pose threat to natural ecosystems and even become pathogenic. Virus networks rely on the transfer of data (encoded in DNA) through a natural process of cell motility. A highly engineered virus may provide efficient communication systems and can produce unexpected mutations presenting fresh ethical challenges.

In the early days of the COVID-19 pandemic, the only way to break the infection chain was masking and social distancing. We present an IoVT application in physical distance monitoring for pandemic situations with a proposed framework consists of three elements: (i) IoT node, (ii) a smartphone app, and (iii) ML-based tools for data analysis and diagnosis. Through the smartphone apps, the IoVT node monitors health guidelines to display the patient's health conditions. Apps alert the patient to maintain a minimum 6 ft physical distance, which is a basic guideline in controlling virus spread to minimize COVID-19 risk exposure.

IoVT comprises four pillars: data, things, people, processes that are intelligently connected with billions of sensors to distinguish calibration and appraise their conditions. "Process" is instrumental in that the three pillars "people, data, and things" interact with each other to deliver value for bestowing immersive user experience in the connected world of IoVT. With the exact process, connectivity bestows value-added because of the concise data delivered to the appropriate person at the appropriate time in the appropriate manner. At the epicenter of the Internet paradigm "Data" is the foundation. Thus, the 3rd generation "Internet for the people" became the 4th generation "Internet about things for people benefit." "Things" include physical devices, sensors, and machines that communicate with each other and to the Internet. "Things" sense data to help machines and people in the IoVT empowering valuable decisions. In summary, the four pillars of IoVT advance open innovation to entrust user experience. When the Internet progresses toward IoVT, people will be connected invaluable manner and transform industries as well as our lives.

### 3.2.2. Trust in IoVT/QmRNA

The theories of QM are so abstract, complex, and philosophical, it is not plausible to find applications outside the realm of quantum theory. Quantum cognition is the application of QM to understand human cognition and psychology. With respect to the quantum vaccine, one needs to understand how trust develops, and how human cognition is affected by trust, specifically in pandemic crisis environments. Trust is ubiquitous in all cognition, and the goal is to comprehend how trust works so that one can better understand what happens when trust breaks down. In quantum mechanics, a context does not directly causes a different result but explains a complex relationship with other decisions that one is making concurrently to decide whether to trust or not. There is a general distrust of QI and comprehending how to generate consequential trust-based interactions between humans and QC will be fundamental to society's success.

Leveraging cutting-edge technology, like mRNA, as a differentiation to bestow up to 95% efficacy in vaccine discovery and development has become the choice of regulators, such as FDA and EMA, hospitals, and healthcare facilities. Trust management should be ethically principled, respectful, and dependable. Confidence management initiatives incorporating innovative application of mRNA technology add patient value. Well-trained

team members seamlessly integrated with mRNA technology the spotlight the following overlooked environmental needs: (i) infection control, (ii) quality assurance, and (iii) regulatory compliance.

### 3.2.3. IoVT/mRNA Security, Privacy, Risk

Every wave of cutting-edge technology, including IoT approbation and enactment has its daunting challenges. Implementing and managing IoVT poses challenges, specifically issues of privacy, security, and risk. The IoVT system is dependent on communications, sensors, cloud, storage, and software programs intensifying concerns about privacy, security, and risk. Such concerns prevail still and are not resolved around privacy and security, creating risk within an increasingly IoVT connected world. As per a recent survey by researchers [77], the top hindrances to IoT growth are privacy and security concerns. Scholars [58] listed that the main security challenges in IoT are (i) access control, (ii) privacy, (iii) policy exaction, (iv) trust, (v) mobile and middleware security, (vi) confidentiality, and (vii) authentication. The IoVT vulnerabilities can be mitigated through security best practices.

To address the security challenges, stakeholders need to organize their thinking in the following manner: (a) security at the design stage, (b)security management to be enhanced, (c) security features built on proven practices, (d) security prioritization, (e) IoVT transparency, and (f) connectivity carefully implemented. As IoT devices proliferate into uncontrolled, complex, and unfavorable environments, securing IoT systems poses unique challenges.

For example, since numerous devices endeavor points of entry within an IoT system, device authorization and authentication are captious for securing IoT systems. Prior to access gateways, and applications, devices ought to establish their identity. Moreover, it should be noted that many IoT devices fall by using weak password authentication. IoT Platforms should provide device authorization services to resolve which apps, services, or resources that each device has access to throughout the system.

The following are the common perceptions and safety profile of mRNA vaccines: (i) COVID-19 mRNA vaccines have been rigorously tested for safety before being authorized by regulators; (ii) mRNA technology is a novel concept, and for more than two decades, researchers have been pursuing it; (iii) mRNA vaccines do not carry a risk of causing disease in the vaccinated person and do not contain a live virus; and (iv) mRNA does not affect or interact with a person's DNA.

### 3.3. Relevant Theories Supporting QBIoVT

In this study, the relevant theories supporting the quantum phenomena are explained to justify the construction of the theoretical framework drawing connections to QBIoVT system applications for vaccine discovery. Since the theoretical framework is defined as the set of existing theories, models, concepts, and relevant definitions that are used in a specific field of study, we have structured the theoretical foundation based on the following theories as the evidence to support and hold a QBIoVT ecosystem:

1.  IoT Interface theory provides the interaction between the thing, service, and user for which it is formalized as an interface automaton for QBIoVT applications, where "everything" and services interact with the physical world [78].
2.  The communication theory bestows the process of information, interdisciplinary disciplines of interpersonal communications, psychological paradigm, and philosophical and social dimension related to QBIoVT applications [79].
3.  The quantum systems' theory applied to QBIoVT systems analysis applications. One of the vital mechanisms of systems analysis is systems thinking, enabling to contour systems from a broad perspective rather than specific events in the system. Quantum theory born as an idiosyncratic theory to elucidate the quantum phenomenon. Based on quantum mechanics, the field of quantum computing has quickly become a quantum tool to achieve a precise representation of molecular systems with improved accuracy in drug design and drug discovery applications. Computational complexity

theorists have long hinted that quantum computers will be the key to unraveling the mysteries of nature and advancing vaccine or drug discovery [80,81].

4.  The Theory of Transparency endows fundamentals of transparency where the actions and decisions of industries, like the pharma industry, are open to inquiry [82].

5.  The theory of mathematical decision is at the core of AI, and linear algebra is something AI experts cannot live without. Mathematical modeling enables the identification of promising in silico drug targets. Decision theory is an interdisciplinary approach to arrive at the decisions and is ensconced into problem-solving approaches in AI power [83].

6.  Complexity theory recognizes that organizational phenomena and bestows a cognizance of how systems evolve. QC is so powerful because it makes quantum algorithms possible even when such tasks are complex. These algorithms demonstrate different complexity characteristics than their classical counterparts. To understand what that means, one must review complexity theory, the study of the computational effort required to run an algorithm [84].

7.  The theory of computation deals with computational demur that can be efficaciously resolved through the computation model using a quantum algorithm [85].

8.  Ethical theory considerations inherent to all the phases of the vaccine discovery process. This theory drives the ethics of care based on moral interpersonal relationships and cares as a virtue [86].

## 4. Methodology

After reviewing the methodological requirements of IoVT forensics and suggestions made by the research community, a context-centered methodology [87] (p. 1) is proposed to perform investigations of the IoVT domain.

Based on the present state of IoVT forensics, challenges, and needs of the pandemic situation, the IoVT dimension of confinement is exploited from the perspective of QBIoVT systems characteristics. Evaluation of the proposed methodology in security scenarios proves to be applicable and effective by researchers in future cases. The scarcity of the security scope executed on IoVT sensors or devices added to the subtlety of the data that has captivated the environment for cyberattacks. Thus, digital forensic investigations have motivated to shed significant insights on what transpires. For the first time, a context-centered methodology is commended to accomplish forensic investigations of the QBIoVT paradigm to impel a specific role in the QmRNA vaccine discovery research.

### 4.1. A Context Centered Methodology

Heterogeneity is the most distinguishing feature of the IoVT. The data that is handled in each scenario are awfully specific and distinct situations, such as the ones related to QmRNA vaccine discovery, which involve extremely sensitive information which requires careful investigation. This means a forensic investigation in the IoVT environment for QmRNA vaccine discovery may have no similarities with any other domain. Forensic sciences require standardization while a brand-new paradigm appears, as is the case of the QBIoVT system. A specific methodology cannot be designed to model all IoT devices, as it will fail to satisfy the requirements. Under such circumstances, an appropriate methodology is imperative. Hence, context-centered methodology [87] (p. 7) is proposed to address specific contexts related to the QmRNA vaccine discovery.

### 4.2. The Topology for the Context Centered Methodology

The methodology can be modeled to work in the IoVT-based operating systems to execute a definitive role in an IoVT network. Example: The Microsoft Windows 10 IoVT Core operating system is selected to design a context-centered methodology that offers many functionalities. This operating system requires to be installed on a device with adequate computational power to execute complex applications to manage the data exchange in the network. This flexibility is found in multiple IoVT scenarios. A user-friendly interface

is essential to interact directly with the IoVT system which allows applications to show information giving the user the option to control the functionality.

For the context-centered methodology, the IoVT topology model, as shown in Figure 2, is shaped for the Windows IoVT core characteristics. The device most relevant in the IoVT network is called the "central node", which executes the operating system. Other embedded systems also perform simpler actions which can be part of the topology as well. The central node implements the applications that provide the functionality to the system and receives information from the sensors as well as sends input to the actuators to perform an action. From the forensic perspective, the central node is the primary source of evidence, since it is the one that stores the information, and the interchange of data in the network passes through it. The "central node" with the Windows 10 IoVT Core operating system is a single-board computer.

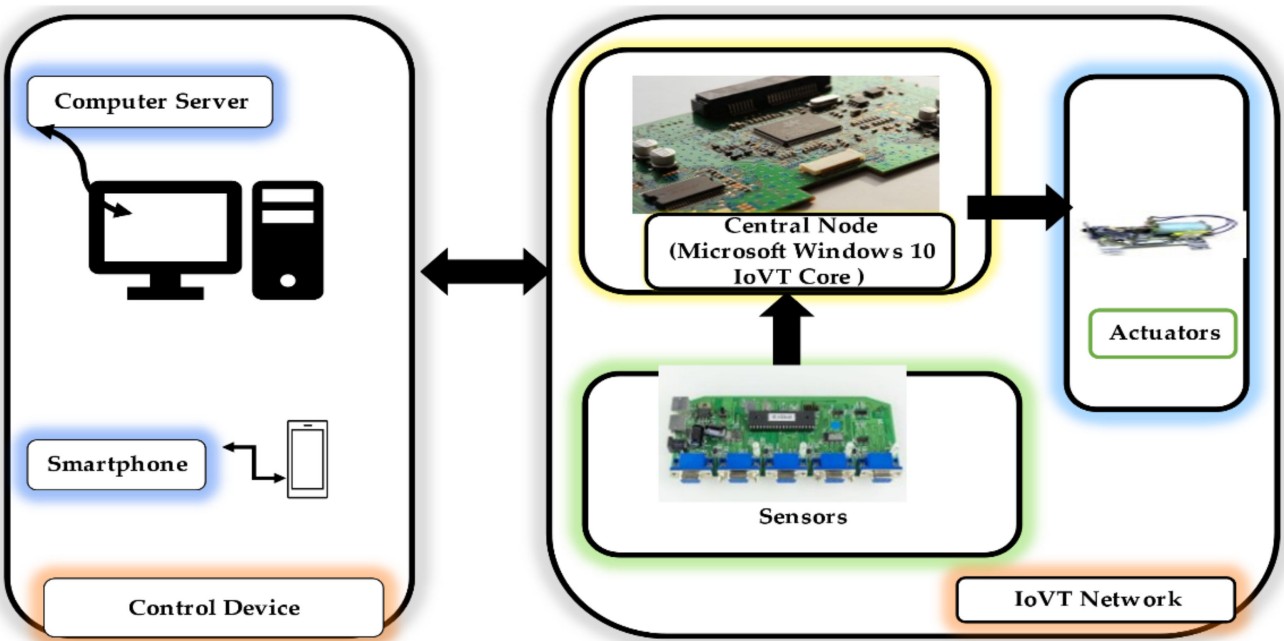

**Figure 2.** Topology for which the methodology is built.

Sensors are designed to carry out simple tasks due to their limited computational capacities and collect information regarding the state of variables. Actuators execute an action, and their characteristics are the same as the sensors. The function of the control device is to interact with the IoVT ecosystem through the services offered by the central node. The main purpose is to monitor the state of the system or send orders to the central node.

This methodology is centered to bestow practical instructions for investigations. The phases that comprise the methodology are the following: (i) identification—the process of determining which devices existing in the scenario to contain relevant data for the investigation; (ii) acquisition—describes the operation that involves the creation of the forensic image of the devices; (iii)analysis—details the inspection of the data contained in each one of the IoVT devices marked, (iv) evaluation—a procedure to group all the data collected from the different devices and conclude how it fits into the environment as a whole.

### 4.3. Research Methodology Motivation

The motivation to design a new methodology to perform forensic investigations in the IoVT environment because it influences the evaluation process significantly which calls for a new approach associated with diversity of sensors, connectivity options, computational capacities, cloud interactions, and physical access.

### 4.3.1. Diversity of Sensors, Devices, and Systems

In contradiction to conventional forensics, IoT objects (devices, and systems) are designed to perform diverse tasks in disparate and specific contexts such as healthcare, smart homes, smart cities, and critical environments. Therefore, the multiplicity of IoT devices and systems is immense. Furthermore, most of them have similarities with desktop and mobile ones. Hence, new propositions on how to accomplish the analysis are necessary.

### 4.3.2. Connectivity

IoVT topographies comprise many devices that interact with each other to perform various actions.

Traditional forensics is egregious to encounter examinations involving multiple devices. Example: a master plan in which there are sensors, actuators, and a central node. The sensor is transmitting data to the central node, which interprets it and communicates (or not) to the relevant actuator to perform an action.

### 4.3.3. Computational Capacities

IoVT devices are designed to exchange information with each other rather than perform complex tasks. However, their computational power, storage capacity, dedicated memory is low bestowing a short lifetime. Therefore, the IoVT methodology requires an approach that allows the exchange of information properly, unlike traditional forensics.

### 4.3.4. Interaction with the Cloud

Because the limited computational capacities of devices are compensated with the usage of the cloud, the IoVT applications in the cloud environment issue is a serious one. Consequently, interaction with the cloud must be considered proactively designing a forensic methodology in the IoVT environment.

### 4.3.5. Physical Access

IoVT devices have such a compact size enabling them to install in small places or be embedded into other objects. An industrial device can be inside the machine that it operates. This means in specific situations; the image of the storage must be obtained by following a live forensic acquisition process.

### 4.3.6. Battery Life

IoVT device installation at some locations is not suitable for connectivity to the electricity supply. Therefore, the use of batteries is a power source influencing the forensic examination if the investigator needs to perform an analysis or live forensic acquisition. Running out of battery power causes an alteration of the data stored on it during the rebooting process, thus influencing the evidence.

## 5. QBIoVT Paradigm

### 5.1. Quantum Biotech (QB)

mRNA medicines are neither small molecules nor biologics which were the genesis of the biotech industry. Biotech, specifically mRNA, exploits bio-molecular and cellular processes to cultivate technologies for vaccine discovery. There are so many things one can do if one can simply create cells that produce proteins from mRNA in the body. The potential of mRNA technologies to transform areas of medicine, including infectious diseases, is immense. mRNA-based vaccines possess the promise to revolutionize the field to offer novel vaccine compositions. QB is based on quantum technology and biology. Distinguishing the game-changing nature of quantum technologies, pharma companies are assembling quantum task forces to overcome the scaling limitations of classical computational methods enabling numerical solutions to tackle the complexity of the molecular systems.

### 5.1.1. QB Building Blocks

The pharma companies are primed to become the beneficiaries of the QB, specifically the QmRNA technology platform. The QB building blocks, as explained below, ensure seamless automated integration of QC, QI, mRNA, Massive Data Analytics (MDA) [41], and IoVT, to scale and assuage an ever-increasing research demand in QmRNA vaccine discovery:

1. QC is a revolutionary technology that has the potential to offer a fundamentally different form of computation and uses three basic properties of quantum physics: superposition, entanglement, and interference. The basic QC characteristic is the capacity to cipher multiple states concurrently and is essential in the era of massive data to process an enormous volume of data. Quantum hardware continues to improve in both quantum bit (qubit) quality, and quantum volume to affect the QVD. Fault-tolerant QC systems are not commercially available yet. NISQ [88] (p. 137) devices and variational algorithms are allowing the approbation and commercialization of quantum chemistry.

2. AI is an ideal candidate for QC facilitating QI with immense computational power for solving optimization problems. A quantum optimization algorithm can integrate all possibilities and produce potential best results and promises to be exponentially quicker in QmRNA vaccine discovery. At an atomic level, QC simulates nature, therefore could identify new materials or chemical compounds for QmRNA vaccine discovery in a few days.

3. Currently, researchers are intrigued between virus and computing similarities. The IoVT would use certain kinds of viruses, which researchers believe make compelling sensor networks. The pathogens share a kinship with components of typical computer IoT devices and viruses to be considered as a living organism of IoT device called IoVT that could play a crucial role in the cold chain storage required to keep the vaccines intact.

4. MDA is cardinal to harness the capacity and dynamism of massive data. Using quantum tools and analytical methods, researchers can undertake analysis of complex data readily to make informed decisions and generate valuable insights. Data can be processed by the software platform, which leverages IoVT, QML, or QI capabilities.

5. For decades, the pharma industry has been operating with a vaccine discovery model that has a low probability of success. Fewer than 10% of promising vaccines that begin the clinical process end up making it to market and that figure has remained constant for years. The convergence of QC and AI facilitates QI that endows great tools to tackle the challenges in vaccine discovery. QI enables major breakthroughs in predictive modeling and harness analytics learning cycle bestowing meaningful research insights that were otherwise unattainable. QI is an ideal solution to improve the notoriously vexing problem and inefficient process of vaccine discovery. It is a significant milestone that assists scientists in every single step to vaccine discovery because it is faster, cheaper, and more precise.

6. The enormous potential of QC and QmRNA technology platforms can function very much like Quantum Operating System (QOS). With an appropriate talent pool, QoS can be designed in such a way that it can plug and play interchangeably with various software programs, as shown in Figure 3.

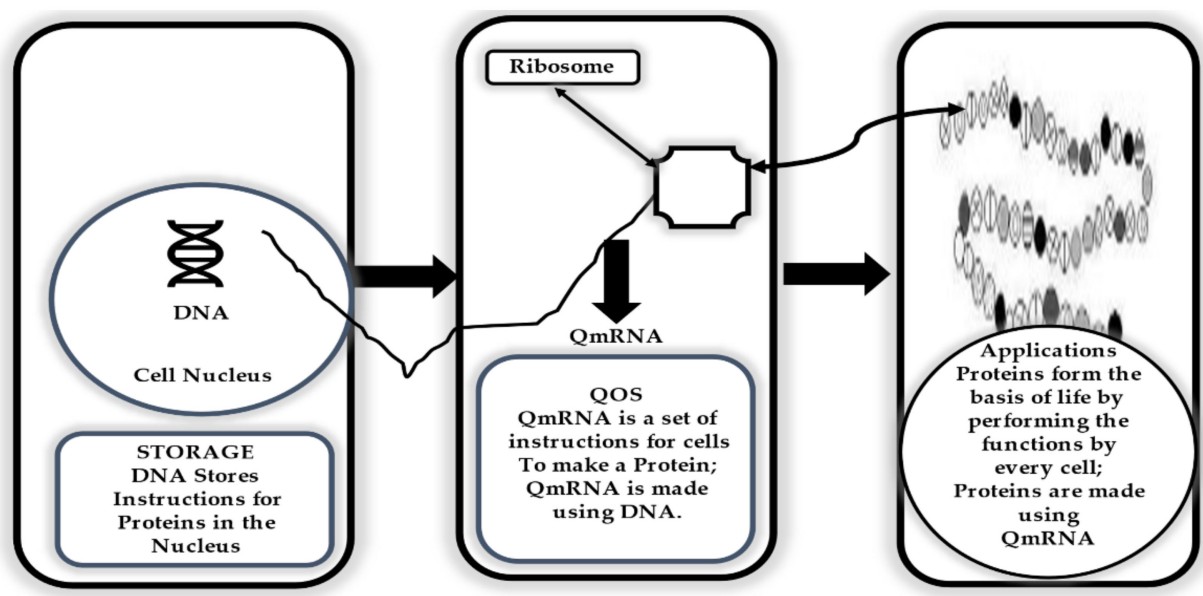

**Figure 3.** Quantum operating system (QOS) for mRNA science and technology platform.

To achieve the QOS mission, the talent pool should be organized around key disciplines like quantum technologies, mRNA biology, protein engineering, chemistry, bioinformatics, IoVT technologies, and the relevant software expertise with quantum language coding. It is vital to note that the QOS should be designed to alter from one potential mRNA medicine to another via the coding domain—the genetic code that informs (instructions) ribosomes to make protein offering investigational mRNA medicines.

### 5.1.2. QmRNA Research Engine

The priority and goals should be to attain the rational design of the QmRNA vaccine and spurring of the research program. QmRNA vaccine design should contain a sequence design apps that allows researchers to build novel QmRNA sequences using a library of sequence components. Embedded QI algorithms transform amino acid sequences into nucleotide sequences and optimize a sequence for production. The next-generation mRNA and protein design quantum algorithms should use neural networks to execute massive data sets to find emergent relationships to enhance mRNA performance to configure the ideal properties for both QmRNA and formulations, as shown in Figure 4.

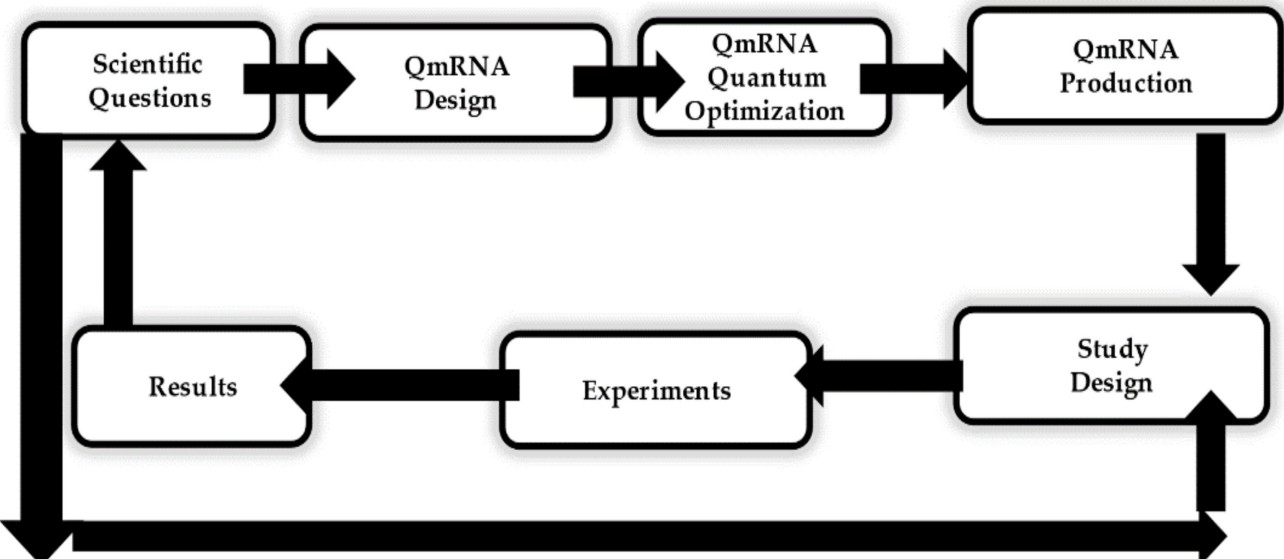

**Figure 4.** Research design quantumization: research mapping.

### 5.1.3. QmRNA Vaccine Technology

The ideal way to deliver mRNA is with a material of highly efficacious delivery that protects it from degradation and facilitates efficient cellular uptake after simple injection. In recent years, the most valuable innovations in mRNA vaccine technology have been in the areas of (i) mRNA sequences engineering; (ii) development of approaches that empower quick, simple, and scaling of CGMP (current good manufacturing practice) [28] (p. 14) production of mRNA; and (iii) development of highly efficient and safe mRNA vaccine delivery materials.

mRNA-based vaccine candidate BNT162b2, has more than 90% efficacy [24]. To advance a QmRNA vaccine, the central strategy must be to enable a quantum infrastructure, built from the ground up to exploit the inherent applicability of mRNA with its QOS software features. It is also essential that several mRNA R&D programs collaborate simultaneously to define a pathway to reduce needless complexity. One must develop core "engines", highly flexible to the unique requirements of scientific and technical endeavors as "research engine" as early R&D engine to drive several mRNA research programs incorporating QOS software and relevant quantum algorithms concurrently from concept to development candidate (DC) nomination and advancing to clinical studies to human proof-of-concept.

### 5.1.4. Enabling Quantum Vaccine Discovery

With the following approach and the application of emerging quantum technologies, the QmRNA vaccine discovery can be feasible in a few days:

1.  To advance new ideas for vaccine candidate discovery to empower a quick supply of preclinical mRNAs for in vivo and in vitro investigations, a research engine is a prerequisite.
2.  Scientists can design mRNAs for research and testing to create new mRNA concepts within days, using hybrid (classical or quantum) systems with proprietary rights. Classical computers use the software like property of mRNA in proprietary, web-based vaccine design. Scientists should also seek mRNAs for a particular protein which is then converted automatically to an initial optimized mRNA sequence. Through a sequence designer module, the mRNA sequence is optimized using proprietary bioinformatics algorithms. Vaccine design uses edge (cloud) [42]-based computational capacity to run various algorithms to design each mRNA sequence. Edge (cloud) system enables on-demand flexible computational capacity, facilitating the research engine to power parallel in-take and design of multiple mRNA sequences. The vaccine design integrates with digital automation platforms directing orders through each phase of mRNA synthesis.
3.  mRNA eon is at the horn of innovation. Early successes and advancements of the mRNA technology platform offer several great opportunities that involve learning the quickest and scaling rapidly, and simultaneously maintaining the highest quality. The best way to achieve such objectives is through the digitization process that includes digital technologies, robotics/automation, analytics, data science, and AI. The following are the benefits of digitization:

    - Quality minimizes human errors by enabling seamless integration, automation, and repeatability wherever required.
    - Fast speed provides large quantities of mRNA across the ecosystem at a quantum speed to permit the acceleration of rational mRNA drug design and to collect, share and analyze data in real-time to make decisions.
    - Significant time savings empower computational chemists and scientists to make new discoveries faster that could lead to quantum mRNA vaccine determination in a few days.
    - Scalability can facilitate an ever-increasing number of mRNA R&D programs within and across methods.

- Cost savings can be attained leveraging across all R&D programs to maximize infrastructure effectiveness.
- Searchability enables the identification of an old receipt (for example) without searching through a vast amount of paper. This means any stored data in a digitized form can be easily searched and receipt software offers to add tags or labels so one can find them later.
- Global access advantage achieved via the availability of internet access empowers one to access vast knowledge over the web globally.

4. To create QmRNA concepts, the above process uses quantum tools and quantum algorithms (QA). Similarly, quantum research engine can integrate proprietary quantum drug design tools and a highly automated facility to enable QmRNA vaccines swiftly through the research stage, from idea to development candidate (DC) nomination, as shown in Figure 5.

5. QM embraces a host of quantum systems solutions that could offer tremendous opportunities to empower drug apotheosis not only to bring the vaccine discovery faster but also can save substantial investment. Quantum computational chemistry, a relevant and effective contraption to determine a molecule that binds to the target protein in the vaccine discovery process, can be achieved with the help of "in silico" tools. QM methods impact addressing pharmacological challenges on the time scale demanded by vaccine-discovery research applications. The selection of the most appropriate method (Molecular Mechanics-MM, or QM, or QM-MM) [61] during vaccine discovery is paramount. It is expected that QM will become a more conspicuous tool in the stockpile of the computational medicinal chemist. Hence, contemporary QM methods will play a more direct role in streamlining the vaccine discovery process.

The application of a QBIoVT system, based on quantum mechanical methods, a valuable contraption for vaccine designers, molecular biologists, and computational chemists to culminate with quantum computational speed to remove 99.99% of deceitful leads in seconds to determine a potential vaccine in a matter of days [27].

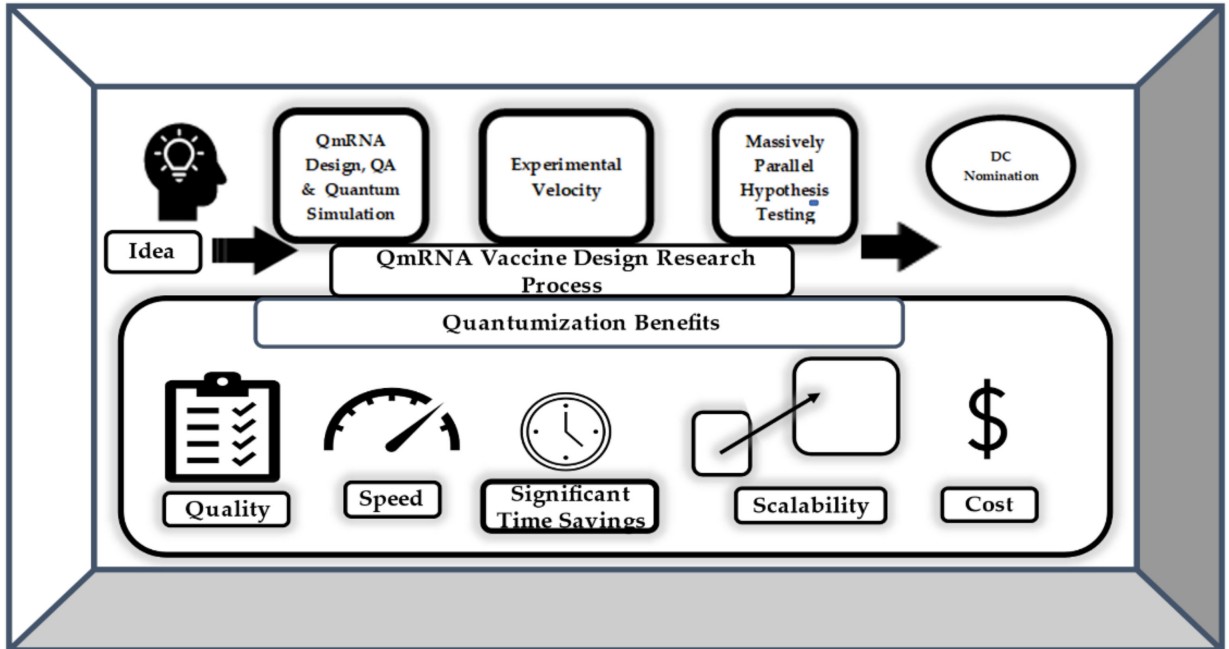

**Figure 5.** QmRNA vaccine design research process.

*5.2. IoVT—Virus as IoT Device*

The simple way to explore and exploit the role of the virus as an IoT device is to compare the microorganisms with existing computerized IoT devices in the field as shown in Figure 6. The characteristics of the virus can be contextualized against standardized digital models in this approach bestowing superior acknowledgment of the species' features.

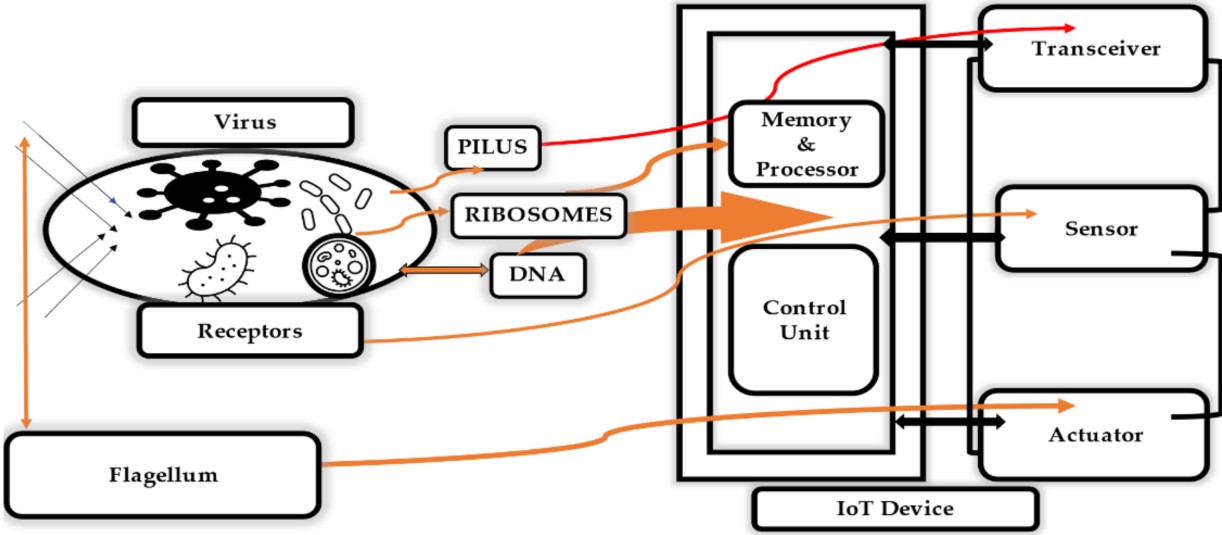

**Figure 6.** Virus as IoT device.

DIY (Do-it-yourself) biology is a germinating biotech social movement in which individuals, communities, and industries investigate life science. The COVID-19 virus is the common, standardized "workhorses" of DIY biology and open-source computing [89], respectively. Hence, both generate a perfect aspirant to be an accessible and cost-efficient tool for worthwhile undertaking and learning. While we can recognize the simplification of both the biological and digital and the universe, the elucidation is designed to raise relevant queries on how viruses could be positioned in the context of human-machine interaction investigations, as explained in Sections 5.2.1–5.2.3 below. These queries may emanate from not only the practical differences in safety in interaction, stability, responsiveness but also ethical.

5.2.1. Sensors and Actuators

The virus can sense a wide range of stimuli, such as electromagnetic fields, light, chemicals, mechanical stress, chemicals, temperature, etc. and may interact through movement using their flagella, as shown in Figure 6, or through the production of proteins, as a response to the stimuli. Virus actuators and sensor's function based on the molecular structure. There can be fundamental differences in their sensitivity, responsiveness, and stability, as opposed to the digital counterparts.

5.2.2. Control Unit, Memory, and Processor

The DNA inside the virus offers encoding of instructions that can be translated into viable functions and provide data storage. It offers as an entity to manage data sets and software conditional expressions units like computer's control unit, memory to store embedded system data, and the execution of software instructions as a processing unit as shown in Figure 6. Generally, there are two forms of DNA present in the virus: (i) genomic DNA containing most instructions for cell functioning, and (ii) circular units called plasmids. Plasmids, in synthetic biology, are often used to formalize various genes into the organism, accomplishing it a versatile contraption for customizing functions and new data storage.

### 5.2.3. Transceiver

A transceiver allows both the transmission and reception of communication, the cellular membrane of a virus that can be considered as a transceiver involving the release and import of molecules as an aspect of signaling pathways for the cells. Furthermore, the bacterial pilus, as shown in Figure 6, is used for the conjugation process between two cells which results in DNA exchange as molecular communication, which forms the basis of virus networks.

### 5.3. A Basic IoVT Architecture

An Industrial IoT (IIoT) platform basically camouflages an identical array of where-withal as the generalized IoT platform but is ameliorated for industrial IoT cases, such as pharma application and the necessities correlated with industrial settings. Generalized IoT platforms are used in various industries and sectors, including pharma, logistics, and many others. Use cases for IoT platforms, such as IoVT, are notably varied and may include smart analytics in a pandemic situation. In this study, a novel IoVT-based model is introduced with an optimized bio-communication interface. The model applies a bio-interface for information collection and transforms it into a composition of an electrical equivalency. Figure 7 shows the basic architecture of the IoVT as applied in vaccine discovery that includes the sensors with the nodal interface, network connectivity, and data storage applications. The architecture stage of an IoVT system includes: (i) sensors/actuators collect data from the environment or object under measurement and turn it into useful data, (ii) Internet gateway, (iii) edge IT system to process the data, (iv) the data center and cloud where the data is stored, and the data analysis is performed.

### IoVT Architectural Layers and Components

IoT device components are identified as it is relevant to comprehend the significance of sensor-based vaccine discovery applications. In this study, we address sensors in both computing (classical and quantum) world and should be noted that the quantum sensors are not prevalent yet. The IoVT architecture comprises four components: sensing, network, data processing, and application layers (as shown in Figure 8) [90], and a brief description of these layers is outlined below:

- Perception layer (sensing layer) is to identify any phenomena in the devices and sensors and obtain data from the real world. This layer consists of several relevant quantum sensors and/or classical sensors depending on the relevant pharma applications. Using multiple sensors for applications is one of the major features of IoVT devices. Sensors (quantum or classical) in IoVT are normally integrated through relevant hubs, a common connection junction for multiple sensors, that accumulate and forward sensor data to the device processing unit. Several transport mechanisms for data flow between sensors and applications used by a sensor hub.
- Transmission (network) layer acts as a communication channel for data exchange, collection of data to other connected sensors and in IoVT sensors; it is implemented by using diverse communication technologies, such as 5G, to facilitate data flow between devices within the network.
- Data processing layer is an integral part of the middleware that includes the data processing unit of IoVT devices and analyzes the data to make the decisions based on the results and the user experience is enhanced by saving the result of the previous analysis. This layer may share data with other connected devices via the transmission layer and given the QC capabilities, the integration of IoVT and QC can address the challenges that hamper the growth of IoVT.
- Application layer, as the user-centric layer, executes relevant tasks for the user data results to the data processing layer to attain IoVT sensor applications.
- Business layer: The culture and methodologies of enterprise structures undergoing a basic shift and entering an unprecedented time in human history. Quantum business modeling is a new archetype of allied discipline performing with integrity, coherence,

and transparency executing a quantum platform for growth drivers. To thrive in the future, pharma industries must adopt a coherence quantum business model to levitate quantum value co-creation to capitalize on the QBIoVT paradigm.

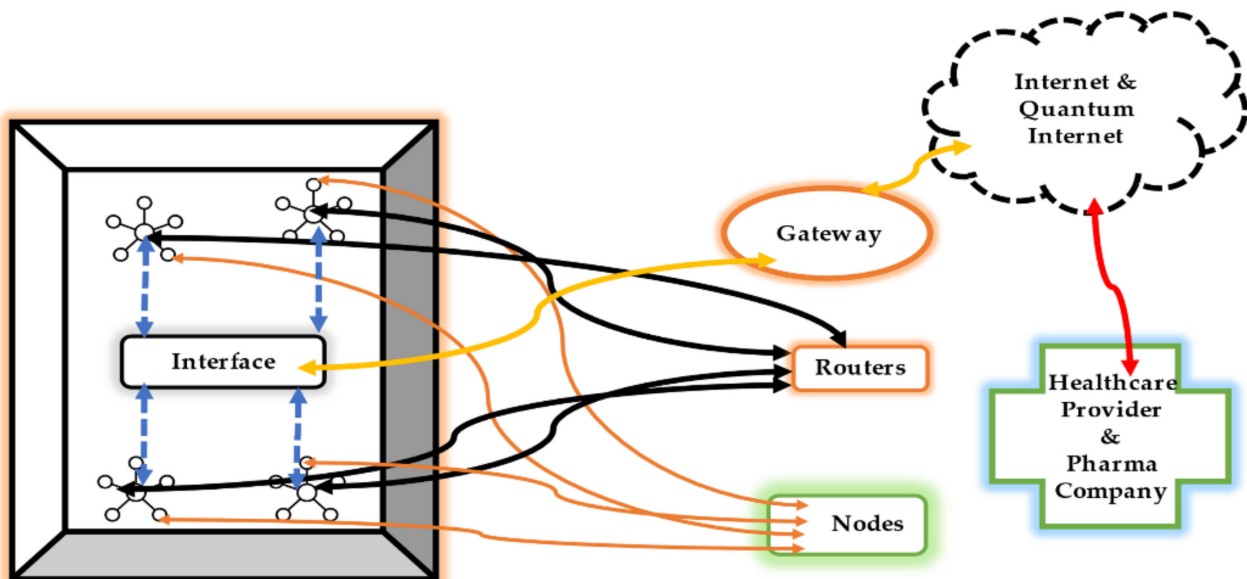

**Figure 7.** A basic IoVT architecture.

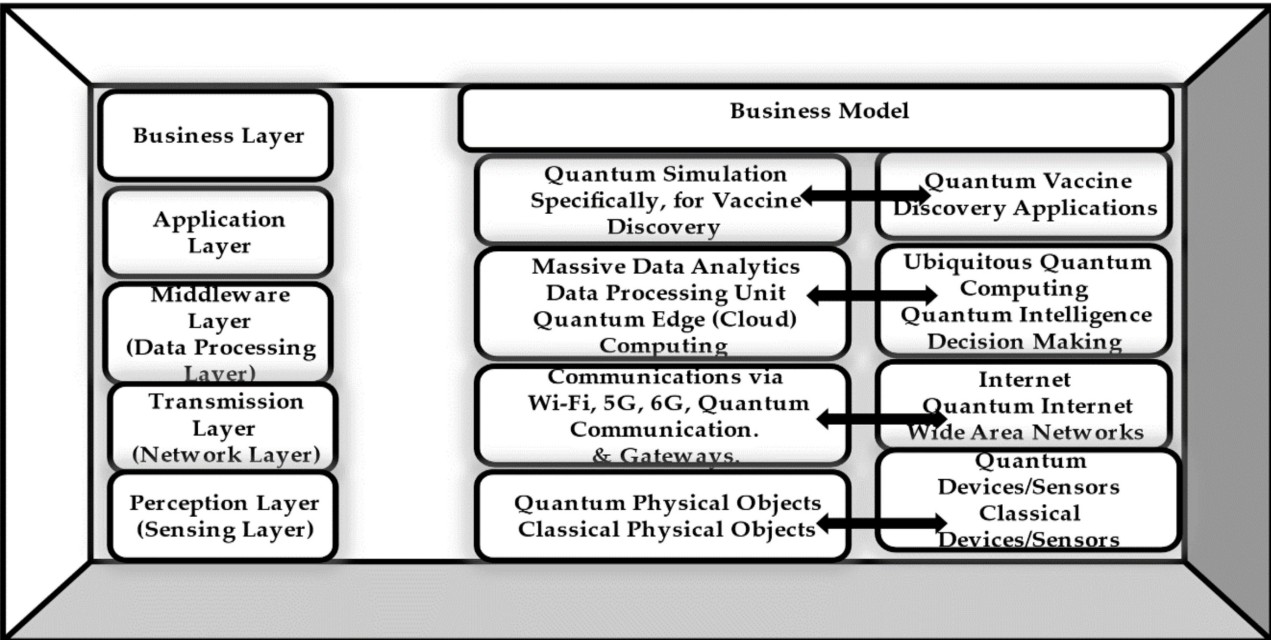

**Figure 8.** QBIoVT architecture layers and components.

*5.4. Merging Quantum Biotech and IoVT to Formalize QBIoVT*

A revolutionary integrated platform of IoVT applications with embedded quantum edge intelligence software technology for various industrial requirements offering the following unique advantages:

1.  Sensors with relevant user interfaces for low latency edge computing performance.
2.  With lower maintenance cost used in vaccine delivery operations with an extended temperature range.
3.  Flexible configurations creating the gateway that fits vaccine production demands.
4.  Scalable performance to reduce the number of transmitted data packages to the edge of the network.
5.  The environmentally friendly platform designed for the lowest power consumption to reduce energy costs and optimize the carbon footprint.

IoVT is a framework in which devices or sensors are connected and work in a synchronized way with the help of software tools that collect data continuously, data integration and analytics, and QI algorithms to inform decisions made about factory operations. The dawn of a quantum era in design automation, called upon the quantum design automation (QDA) [31] industry for more sophisticated tools to help the advanced QI, and IoVT. QDA is to convert digital design into structured quantum science. One of the most amazing things going forward is to see if QDA can bring the same level of rigorous structure to the design of neural networks and the design of quantum algorithms. It is in this space that AI can, and will, have the greatest impact to expedite the QmRNA vaccine discovery process by enabling the power of QI to quickly identify a vaccine. As quantum biotech, specifically QmRNA, quantum intelligence, and IoVT merge, a novel quantum ecosystem called QBIoVT is formalized.

## 6. QBIoVT Priority Areas, Challenges, and Applications

*6.1. Priority Areas and Challenges*

6.1.1. Priority Areas

The topmost priority areas are safety criteria, supply chain, customer experience, asset management, and financial decision-making. First on the priority area is the customer experience improvement. The following are ways the QBIoVT system can enhance the customer experience:

*   Safety criteria execution.
*   Effective distribution management and supply chain.
*   Customer experience continuous monitoring.
*   Customer personalization based on each situation.
*   Based on learning over time with customer interactions, products and services need to upgrade.
*   Purchasing experience through continuous innovation product access.

Asset management and financial decision-making are the second and third priority areas, respectively. The QBIoVT system can play a significant role in financial decision-making by facilitating asset tracking and managing real-time visibility. Customer experience and financial decision (asset management tracking) facilitate a sound foundation of functional areas for practitioners and academic researchers to appraise. Furthermore, the following specific priority areas of QBIoVT needed attention:

*   Quantum prioritization—quantum approbation needs to be informed decisions regarding QC use cases applied to specific pharma enterprise requirements. Prioritizing QC use cases is a captious endeavor. Quantum prioritization empowers pharma professionals to evaluate the following: (i) capacity to deliver quantum technological benefits, (ii) operational preparedness, and (iii) competency to drive unique value co-creation for pharma enterprise.

Quantumization [7–11] building blocks are imperative to ensure seamless automated integration of QB, specifically QmRNA technology platform, MDA, QC, QI, and IoVT, to scale and assuage an ever-increasing demand for QmRNA vaccine discovery research.

- Edge computing [42] will take prominence over cloud computing in the IoVT utilization.
- Better MDA.
- IoVT: Creation of unified IoVT framework for integration, security, and customer-focused priority perspectives.

6.1.2. Challenges

1. There is no specific guidance for mRNA vaccines from the regulatory agencies such as FDA or EMA; they have accepted the approaches proposed by various pharma companies and other relevant organizations to demonstrate that the mRNA vaccines are safe and acceptable for testing in humans. It is an imperative priority, as mRNA products become more prominent in the vaccine domain, to formulate specific regulatory guidance that will delineate requirements to produce and evaluate new mRNA vaccines. Therapeutic challenges include scaling up good manufacturing practice (GMP) production increasing efficacy, constituting regulations, and safety. mRNA vaccine perishability is a major issue. Since QmRNA is emerging, discipline-specific guidance is also required for GMP increasing the efficacy.
2. At present, a quantum challenge is to make qubits less error-prone so that the design of ever-larger circuits can be attained without the challenge that errors will asphyxiate the computation accurately. To counter the challenges of quantum programming, the software developers must comprehend the terminologies, tenants of and exploit solutions that can be applied in quantum computing. The quantum computing daunting challenge is to achieve the scalability into the millions of qubits, a "supremacy number" of logical qubits without errors.
3. QI has the potential to revolutionize the very elucidation of molecular comparison by enabling the development of methods to analyze larger-scale and complex molecules. The following are the four major challenges for the QI future: (i) reinstate repetitive training with faster quantum algorithms, (ii) extract the experience of massive amounts of data into the training process, (iii) acquiesce classical and quantum components to be easily integrated and replaced, and (iv) frame tools to exhaustively analyze advantages stem from quantum algorithm properties.
4. Viruses interact directly with the micro-organisms to bring practical and ethical challenges. These include handling microbes safely and the bio-ethical and biosafety challenges. Ethical considerations and privacy matters related to user data would apply to IoVT systems. Virus networks rely on the transfer of data (encoded in DNA) through a natural process of cell motility. Although the highly engineered viruses may provide efficient communication systems, eventually, they are biological entities, which can produce unexpected mutations presenting fresh ethical challenges.
5. Supply chain, distribution management, and transporting vaccines is a challenge. mRNA vaccines need cold storage during transportation; exposure to sunlight and fluorescent light must be avoided. IoT empowers technological development and support to the vaccine distribution system. The impact of IoT will affect how mRNA vaccine supply chain leaders access information needed to improve the service. The IoT revolution can improve supply chain management by smart connectivity [43].

*6.2. IoVT System for COVID-19 Pandemic*

6.2.1. Performance Management Insights

IoT-based software solutions provide a real-time key performance indicator (KPI) to support increased transparency and performance dialog. The software evaluates device information on overall effectiveness and quality through IoT connectivity. Such digital tools monitor improvement actions and transmit alerts to relevant personnel via mobile devices as well as boost significant productivity.

6.2.2. IoVT System Architecture Insights and Applications for Pandemic

In the early stage of the COVID-19 pandemic, there was no designated therapeutics cure or vaccine. The only way to protect from the COVID-19 infection was self-isolation, masking and maintaining the physical distancing at least six feet. The following elements can provide a six feet physical distance, as shown in Figure 9:

1. An IoVT node with sensors placed at various locations for application-specific, effective, low power, easy to use solution for real-time challenges and monitors health parameters, then updates the smartphone applications to display the user health conditions to notify the user to maintain a physical distance of 6 ft, which is a key aspect in controlling the spread of the infection. Sensors are the input providers from the physical world; sensors are also the input providers over a network, and actuators allow exploiting or react as per the input received from sensors. The cloud server with a fuzzy system considers user health conditions to forecast the risk of infection spread in real-time and the risk conditions convey from the virtual zone and bestow updated data for different places. Thus, an IoVT node can provide an early warning system to curb the spread of infectious diseases and can leverage to enable a healthcare system to deal with pandemic outbreaks.

2. A Cloud server with QML tools is utilized for MDA and diagnosis. Data communication (as an input) is done via a gateway which shall further be transmitted to the quantum edge (cloud) gateway. In the massive data warehouse, filtration of data is distilled. ML and QML are used to create system models based on the needs and accepted data. MDA is used for visualization of performance comparative study and outcomes. Infrared (IR) sensors are used in public places, including toilets, for the automatic operation of water supply, doors, and windows. IR thermometers are used to check the body temperature to identify infection of the people and facial recognition by using the optical camera at all entry points of all public places including airport gates, bus stations, malls, and railway stations. Sensors are installed to monitor the body temperature, automatic operation of windows, lifts, escalators, doors, water supply control, toilets, online conference to avoid direct contact with the physical world and human's interaction.

3. Smartphone Application: Smartphone technology offers significant applications in the current COVID-19 pandemic situation. With the introduction of the 5G mobile system, smartphone technology further evolves to play a key role in future pandemics, and many other applications of health care. Smartphone applications enabled with IoT use information such as Geographic Information System (GIS) and Global Positioning System (GPS) for monitoring purposes to increase the chance of detecting infected people. Example scenario: The Pandemic-Safe framework communication (4G, 5G, or Wi-Fi between the IoVT node and cloud server) can help in minimizing the exposure risk to COVID-19 as shown in Table 5 and Figure 9.

**Table 5.** Sensors/devices and applications.

| Serial No. | Sensors/Devices | Applications |
| --- | --- | --- |
| 1 | IR Sensor | Power switches; water supply control; lift, doors, windows operation. |
| 2 | IR Thermometers | Body temperature measurement; thermal imaging. |
| 3 | Smart Watch | Heart rate detection |
| 4 | Optical/IP Camera | Virtual meetings, facial recognition |

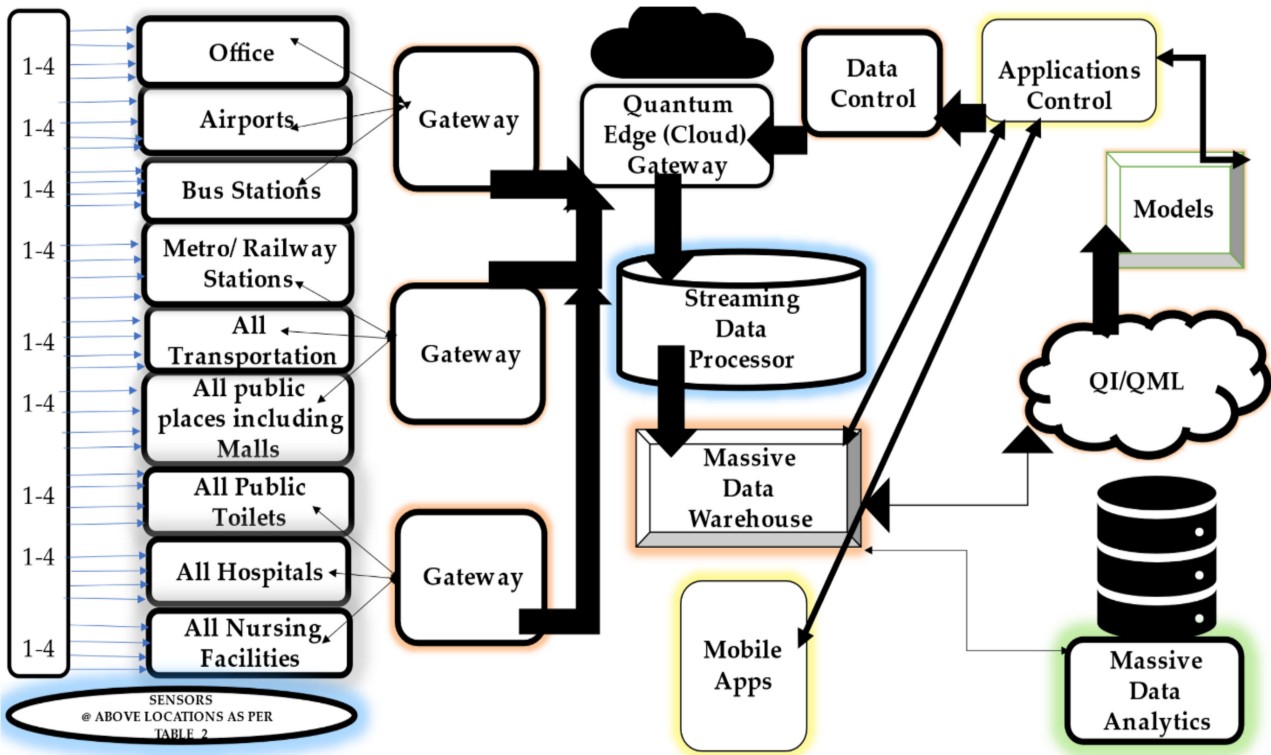

**Figure 9.** IoT applications in the COVID-19 pandemic.

IoVT system plays an instrumental role in the pandemic situation for monitoring of infected persons. All high-risk infected persons can be tracked and traced conveniently using the internet-based network. This system can also be used for biometric measurements, and the workload of medical professionals can decrease as well as achieve cost savings, as shown in Figure 10.

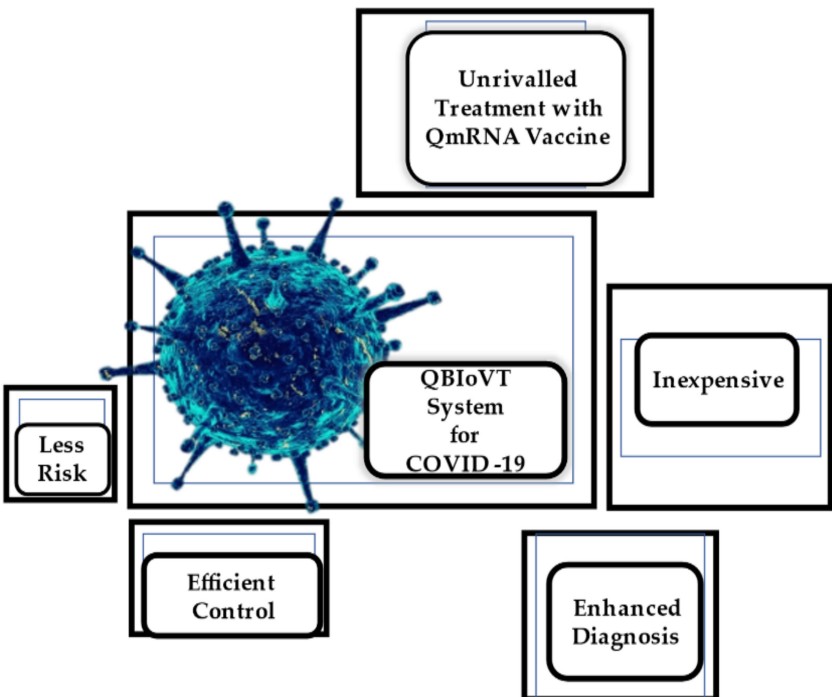

**Figure 10.** Merits of the IoVT technology for COVID-19 pandemic.

## 7. Towards a Novel Theoretical Framework

Literature review, in Section 2, verified that there are not any theories advanced yet within the QBIoVT system for QmRNA vaccine discovery. In our view, a theory constitutes avant-garde benefaction to a domain in the following manner:

- A theory's core phenomenon does not cowl preceding theories.
- A theory's ingenuity may originate because of important renovation it evokes to a current theory explaining the boundary of the theory precisely.
- A theory may be recognized as avant-garde because of its composition to entrust well-known nucleus phenomenon in new contrivances.

The QBIoVT theoretical framework developed in this article falls under a theory's core phenomena that have not been blanketed with the aid of using earlier theories.

Based on a set of applicable theories, concepts, QBIoVT priority areas, applications, challenges, stated in Sections 3–6, the theoretical framework was developed and classified into cyberspace and physical space based on the following perspectives: (i) quantum technology innovation and approbation of QBIoVT system; (ii) QBIoVT stakeholders—as stimulated by the interest in the emerging QBIoVT system; (iii) utilization attention of QBIoVT priority areas, challenges, applications and safety, privacy, risk security issues; and (iv) usage change as human beings gain confidence and trust in QBIoVT system.

As associated with the alternative components of the framework, the hyperlink among QI and IoT is remarkably close; that is witnessed by the term "Internet of Abilities" (IoA). The IoA is the "abilities of human capacity and AI capability are synergistic, transferable, interconnected. Furthermore, linked to IoA is the concept of the Internet of Virus Things (IoVT) four fundamental pillars: "people", "data", "processes", and "things", as mentioned within the theoretical basis segment of the study, in which the people and their skills are expressed by IoA as its foundation. IoVT has a broader scope in pandemic situations, hence, epitomizes the core enabling technology of the QBIoVT system framework, and it is ensuing expansions.

Regarding the functioning of the framework, it is observed that the most used enabling technologies are related to IoVT, QI, mRNA empowering the powerful open innovation (OI) processes that are fundamental to QmRNA vaccine discovery. MDA merits a special mention to offer the background information the pharma industry wishes for its decision-making processes. In addition, MDA combined with OI processes and the use of enabling digital platforms that lead to value co-creation. In this framework, people are thus at the very core because they enable interactions, which in turn are the basis for both OI and value co-creation processes. This insight is of exceptional interest which is to be studied in-depth after further development of such individuals.

It is vital to recognize the enabling technologies via the intervention of mRNA technology platform, QI, IoVT can form a virtual environment for daily life surroundings. In this context, QBIoVT represents the real hyperlink between virtual space and physical space, specifically, it functions as a cyber-physical systems (CPS) domain. CPS is an evolution of cyberspace that offers the IoVT objects positioned at the epicenter of the framework as a hyperlink between physical space and cyberspace.

The QBIoVT priority areas will vary for diverse quantum vaccines and therapeutics relying upon patient experience, asset management, and financial decision-making. Stakeholders play an important role in theory formulation; hence, scholars need to pay attention and decide which stakeholders must be included in a theory. In the QIS and/or IS discipline, most theoretical frameworks are process frameworks or variance, consequently, scholars advocate due to the technology adoption, the theoretical development must be from a process perspective.

QBIoVT challenges (safety, security, privacy, confidence, trust, risk) are taken into consideration as an essential factor of this study. Security and safety measures are paramount to gain confidence and acceptance of trust in the QBIoVT services for pandemic situations. Confidence advent and conviction management play a prominent role in the QBIoVT system for reliable data fusion, superior privacy, context-conscious intelligence for various

services, and information security. Also, in this study, additional contributions (stated below) on confidence and conviction management have been established for the development of the QBIoVT system theoretical framework. In our view, confidence and conviction creation and sustainability must be a proactive approach in all phases of QBIoVT system adoption and execution.

Furthermore, scholars have conducted an in-depth survey of trust analysis and computation models for IoT concluding with guidance for trust computation research. Based on such a prior survey and the insights, this study gave credence using those survey judgments while developing the novel theoretical framework. While confidence and conviction management, security, safety, and privacy are essential to the QBIoVT system adoption and execution success, privacy, safety, and security are the harbinger to the creation and sustainability of trust management. Thus, the above insights concerning the developed framework assist to analyze QBIoVT applications and services.

The development of the proposed theoretical framework is focused also on the building blocks and practices, as shown, are the following: (i) the design of QBIoVT architecture to exploit the priority areas, challenges, and opportunities; (ii) cooperation and participation among the stakeholders within the QBIoVT value chain; (iii) interactive actions between the quantum biotech system, and IoVT applications to achieve optimum performance within the QmRNA vaccine discovery. Based on the building blocks, as shown in Figure 11, QBIoVT theoretical framework is constructed, and the framework embraces the stakeholder's interactive work.

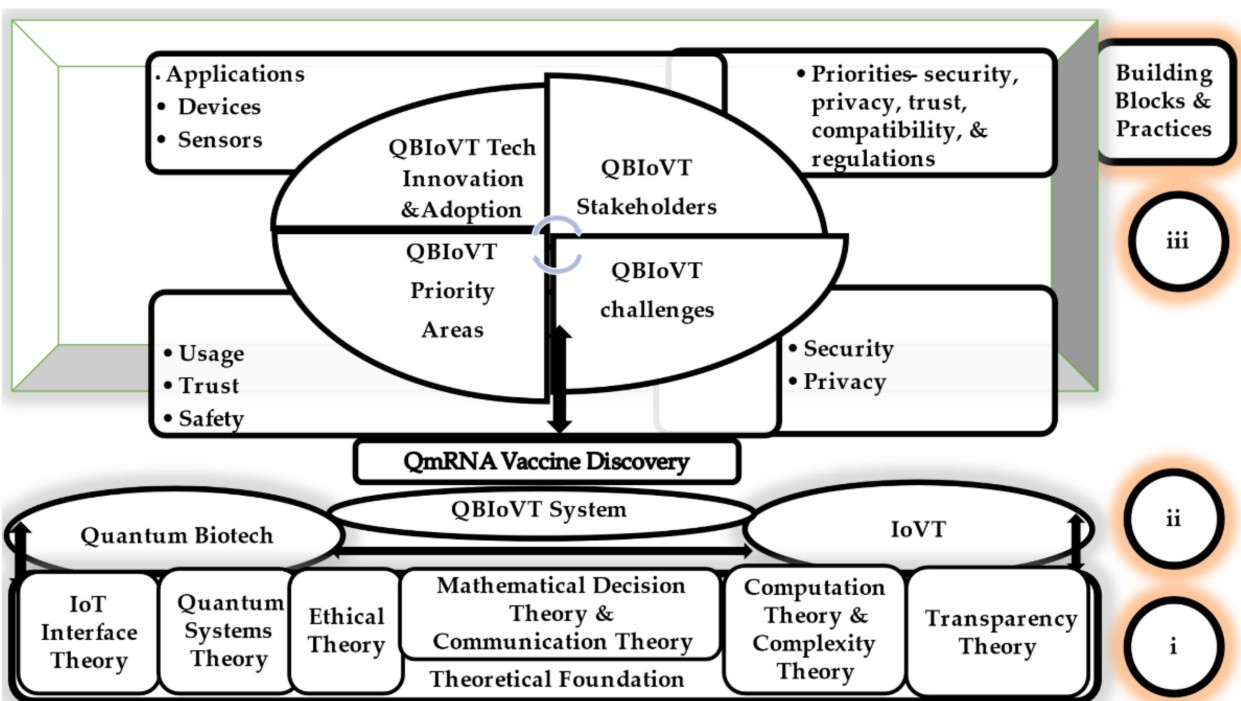

**Figure 11.** A novel theoretical framework.

This framework may be used as an education model and guide for pharma and health industry stakeholders inclusive of suppliers, providers, customers, and decision-makers who are seeking QBIoVT systems. Thus, the primary contribution of our studies is the advent of a novel theoretical framework on the QBIoVT system approbation and execution which is non-existent within the literature.

The following are additional contributions of the framework additives associated with the pharma industry applications and challenges: (i) QBIoVT system entails people to machines (P2M), machines communicating with machines (M2M), and people to people (P2P). All stakeholders are paramount for the ultimate success of the QBIoVT system.

Garnering this in mind, cogitation of challenges and priority areas governing QBIoVT system adoption and implementation must receive special attention. (ii) It contributes to quantum intelligence, IoVT, and QmRNA knowledge guiding practitioners especially imparting attention to the challenges of security and privacy to attain trust and successful use of QBIoVT system in various pharma and healthcare industries. Since the number of connected devices, sensors, and machines increases exponentially, industry professionals are required to assess the QBIoVT system paradigm challenges and opportunities. (iii) IoVT systems would possibly influence pharma enterprises indicating the business models may need to be redefined. (iv) It bequeaths to the body of knowledge in the explanation and prediction on the theory constituents. Ample scope for testing this theory and constituents of the theory should be alternatively investigated further. (v) It offers a perspective road map to tackle the quantum vaccine discovery and IoVT execution impacts on reducing vaccine research and development cost and improve treatment outcome of the infected patient. (vi) It bestows managerial significance for pharma companies to maintain a strategic competitive advantage by adroitly adopting the following IoVT strategies: (a) the typography of IoVT strategic decision is beneficial for decision-makers to identify the rightness of the IoVT strategy they compose and (b) to better comprehend the characteristics and essence of selected IoVT strategy.

The proposed framework is a refined comprehension concerning the strategic selection of IoVT. By evaluating IoVT techniques of the pharma corporation, we have composed 4 categories of IoVT techniques with respect to IoVT contexts: (i) outmaneuver approach in technology innovation, (ii) loom up approach in technology integration, (iii) outmaneuver approach to attain marketplace success, (iv) loom up approach within the sustainable differentiation. This topography evaluation empowers pharma businesses within the IoVT system commercial enterprise to emerge as extra effective. Hence, we echo the decision for further research to assess emerging contentions associated with the IoVT system from a strategic perspective to analyze the IoVT phenomenon.

## 8. Consolidated Lessons Learned and Future Research Agenda

### 8.1. QBIoVT System Theory Development

A void exists within the literature organizing the theory development and principal improvement on QBIoVT system adoption and execution considering the priority areas, challenges, applications, and enormous possibilities for QmRNA vaccine discovery, especially while humanity is going through a pandemic. Therefore, we recommend researchers pursue further studies within the disciplines associated with QIS, QI, QmRNA, and IoVT to reap the essence of the theory.

### 8.2. QIS Innovations

From a conceptual stance, there are not enough motives to just accept that QIS is qualitatively exceptional from classical information systems (CIS) or truly known as IS [13]. QIS is an emerging discipline and is well-positioned at the intersection of technical, business, social and ethical applications of QIT, QB, quantum operational technology (QOT), and its critical dimensions of IoVT. The QBIoVT system open innovation (OI) is emerging to unharness possibilities QmRNA vaccine discovery. The QBIoVT system emergence harbinger a completely new quantumization dimension [7–11] with impacts that are completely unknown. This synopsis welcomes the possibilities for further research inquiries. We advise that applicable future research work is critical to broadening theories on QIS, QmRNA, QI, and IoVT helping moral, philosophical, and socioeconomic components for pharma industry applications to create authentic value. Thus, the subsequent future studies are recommended: (i) QIS multi-stage exploration, (ii) thematic effect discipline: (a)organizational impacts; (b) effect on quantum technology; (c) influence on individuals; and(d) effect on society.

### 8.3. Managerial Significance of IoVT

The following instructions can also additionally assist pharma companies to compete in this emerging, fast-growing, but much less known, IoVT discipline: (i) to embody an outmaneuvering method to the inner advancement of exploitative competencies. To date, such outmaneuver IoVT method are not accomplished with the aid of using pharma companies via suitable development of critical helping competencies, and (ii) the inner and outside enterprise facts sharing have disparate influences on the respective helping competencies of IoVT strategies.

### 8.4. IoVT Technology for Pandemic

The IoVT is a large ubiquitous community of sensors globally to uncover viruses that could empower an early warning system to restrain the escalation of pandemics. The worldwide cooperation might require a foundation to undertake and execute the IoVT network that could place as an advantage for humanity. To gain this paramount intention compels a brand-new mindset. We opine the sort of useful intention to be the "holy grail" of IoVT possibility for the future. Hence, a network of virus-detection sensors together with facial recognition and location, digital digicam surveillance to track, identify, and reveal people that can also additionally had been infected via COVID-19 must be followed and executed. Although this concept can also additionally mean totalitarian rule to many, ultimately, leveraging the IoVT era can be the maximum applicable manner to keep away from pandemics globally.

### 8.5. mRNA

mRNA technologies have the capability to revolutionize areas of medicine and higher know-how of excellent traits principal translation efficacy and the significance of mRNA delivery influencing a brand-new generation of investment in vaccine discovery activities. Enduring enhancements and optimization closer to growing the subsequent generation of mRNA-based vaccines are certainly happening. Sequence optimization within the coding regions and untranslated regions (UTRs) [28] of mRNA facilitate greater efficiency and balance to bring about better fructification of the desired antigen, greater favorable for therapeutic and vaccine index [24] (p. 3). We in short spotlight 3 interrelated subjects that, if better understood, ought to propel the sphere further: (i) variations in mRNA preparation, (ii) differences between animal models and humans, and (iii) mechanisms of immunogenicity of mRNA vaccines.

Researchers envision and prescient a future well beyond vaccines and the subsequent act for mRNA will be larger than COVID-19 vaccines, enabling reasonably priced gene fixes for most cancers and perhaps even HIV cure.

### 8.6. Quantum Computing (QC)

IBM envisages a commercial fault-tolerant QC system in the subsequent decade. Researchers completed higher quantum simulation accuracy of molecules to design new materials, with fewer qubits by executing the behavior of electrons directly into a Hamiltonian concept [88] (p. 137) which has the capability for quicker vaccine discovery. Currently, superconducting qubits technology has the capability for QC. This does not mean other technologies for making qubits will not find a place to succeed. Trapped-ion QC can also additionally discover an area however the avenue to development is a mile longer process. QC applications may require "hardware-aware" so that software tricks can be made optimal use. The varieties of qubits best suited for simulating the behavior of new material are probably disparate for financial portfolio optimization. Researchers contend that quantum computers will work in concert with classical computers, constructing on each other's strengths. IBM makes use of the open-source community support and mobilizes developers to democratize access to quantum technology and identifies the subsequent segments for laying the inspiration for the future: (i) producing high-overall performance quantum circuits at the bottom tiers and groundbreaking algorithms based on these circuits to be

developed, (ii) these algorithms ought to be carried out to killer applications in the real world. Microsoft in 2021 introduced, the world's first full-stack, Azure Quantum public cloud ecosystem for quantum solutions for developers, researchers, systems integrators, and clients to learn and construct solutions using tools within the public cloud [91–93].

### 8.7. Quantum Intelligence (QI)

QI, an emerging powerful technology, is predicted to have a towering impact on the biopharma industry specifically in QmRNA vaccine discovery with the aid of using and extensively improving the quality and accelerating the massive data analysis process. Furthermore, QI bestows compelling tools for investigating complex systems to steer and encourage cutting-edge quantum technology companies to enter the brand-new vaccine and therapeutics marketplace competing with pharma organizations.

### 8.8. Quantum Simulation in the Context of QmRNA Vaccine Discovery

QM cradles several quantum systems solutions that would offer tremendous possibilities to empower vaccine apotheosis not only delivering the quantum vaccine discovery quicker but also saving substantial investment. "in silico" tools have made quantum computational chemistry an applicable tool to determine a molecule that unites the target protein in vaccine discovery research. The choice of the maximum suitable method (Quantum Molecular Mechanics-QMM) [93] during quantum vaccine discovery is paramount. It is predicted that QM will become a greater conspicuous tool within the stockpile of the computational medicinal chemist and current QM methods will play a greater direct function in streamlining the quantum vaccine discovery process.

QBIoVT system, based on the quantum mechanical methods, a precious contraption to culminate with quantum computational pace to eliminate 99.99% of deceitful leads in seconds to ordain a promising QmRNA vaccine candidate in a matter of days.

### 8.9. Quantumization for QmRNA Vaccine Discovery

Quantumization is an emerging phenomenon that comprises building blocks to empower seamless automated integration of mRNA, QC, QI, MDA, and IoVT, to scale and assuage an ever-growing demand for QmRNA vaccine discovery research. Furthermore, the self-ancillary consequences of QmRNA are a contradictory saber for quantum vaccine efficacy.

## 9. Conclusions

This study completed the goals as mentioned in the introduction, Section 1.3. To fill the void with new knowledge and understanding concerning idea improvement on QmRNA vaccine discovery, this examination advanced a novel QBIoVT system theoretical framework. Furthermore, so far, the quantum system application for the QmRNA vaccine discovery has not been covered or protected through earlier theories. Therefore, primarily based on the theoretical framework advanced in this paper, research directions may be drawn adducing adjustments within the present idea or new theories.

Regarding the findings, proof of the QBIoVT paradigm, which include dialogue on architecture, priority areas, applications, challenges, and effect of the QmRNA vaccine discovery, has been identified. It is essential that the present vaccine discovery system need to be transformed through making use of powerful quantum technology and tools to reap the better efficacy and expedite the process in a few days.

Stankovic [94] states that significant studies are critical to gain on the IoT scale, scope, and spectrum of studies. During the COVID-19 pandemic situation, studies suggest that firms are "doubling down" on the IoVT applications. Although the IoVT contemporary evolution is a practicable construct, research corroborates its significant value to the pharmaceutical industry stakeholders.

As IoVT continues to propagate, protocols, standards, and connectivity must ambulate. Concurrently, trust, security and privacy issues must be paramount and take precedence within the approbation and execution of the IoVT system.

Additionally, the massive amount of data managed within the IoVT perspective poses research challenges in the trust, security, and privacy domain.

This research has a few limitations, especially the lack of an empirical test of the theoretical framework proposed, and consequently, further investigation is desired.

Through an understanding of the QBIoVT system theoretical framework proposed in this paper, each researcher and practitioner should experience trust, opportunities, and benefits within the quickly developing IoVT realm.

**Author Contributions:** Conceptualization, P.K.P.; methodology, P.K.P.; validation, P.K.P.; formal analysis, P.K.P.; investigation, P.K.P.; data curation, P.K.P.; writing (original draft preparation), P.K.P.; writing- review and editing, P.K.P.; visualization, P.K.P.; project administration, P.K.P.; Supervision, F.C.-S. Both authors have read and agreed to the published version of the manuscript.

**Funding:** This research received no external funding.

**Acknowledgments:** The authors would like to greatly thank Fundação para a Ciência e Tecnologia (FCT) and C-MAST (Center for Mechanical and Aerospace Science and Technologies), under project UIDB/00151/2020 for their review support.

**Conflicts of Interest:** The authors declare no conflict of interest.

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
