# Peer review of "Quantum Biotech and Internet of Virus Things: Towards a Theoretical Framework"

_asi, doi:10.3390/asi4020027_

Round 1

Reviewer 1 Report

This article investigated the quantum biotech internet of virus things (QBIoVT) in terms of architectural aspect, priority areas, challenges, Apps, and future directions. The authors presented theoretical framework to describe detection and protection for the COVID19 control. This study is recent, important and adds to the knowledge science. However, the authors must address all of the following concerns accurately.

  • The abstract requires improvement, it is incomprehensive (The authors should explain briefly purpose/significance, method/process and result/conclusion) and dependent (The abstract should be entirely understandable (self-contained text) on its own to a reader who has not read the whole article/associated research).
  • We suggest that the authors should eliminate keywords such as “Internet of Virus Things” and “Theoretical Framework” because these keywords are already found in the article title. It is preferable that they replace them with other words in order to expand the reach of the search.

  • The introduction is comprehensive and contains a lot of valuable information, but it is a little long. It is preferable to reduce the size of the introduction section. Also, the authors' real names should be used instead of a word "authors" (page2-line82, check whole article).

  • Section 1.2 (Significant Contribution) should improve, more detail is required because this section demonstrates to the reader the importance of this research.

  • Some acronyms require definition before they are used, such as mRNA, IOE, FDA, EMA, ... etc.

  • The authors referred to a set of databases (page4-lines167-169), they claimed to be "double blind", in fact not all of them were "double blind", the authors should write the information accurately.

  • In section 3.2.3 (IoVT/QmRNA Security, Privacy, Risk (page11)), the authors should provide some real-world examples of main security challenges.

  • The methodology requires some improvement.

  • The authors noted the benefits of digitization (page 16). what about Searchability and availability.

  • Section 9 requires some improvements. Although the authors pointed out the limitations of this study, they should also describe the findings of this study clearly and briefly.

  • Figures and tables: The authors drew the figures and tables well. However, there are some issues. Figure 3 requires minimization. In Table 1.c and 1.d, some references do not contain a year such as [60] and [78]. In Table 2, the lines should be black color. In Figure 8, why are different colors of the arrow lines used? Figure 10 requires zooming in to be clearer.
  • In general, this article is very long and requires reducing the size by removing unnecessary, duplicate sentences or improving writing.
  • Proofreading: This article requires extensive proofreading to address English writing issues. There are many typos, grammar, and spelling issues that need extensive scrutiny. Also, some sentences, as well as paragraphs, are long that make the meaning unclear, the authors should address this issue. In particular, the authors should paraphrase many of the sentences (check the entire article) to make it distinct from the relevant papers and add necessary references if required. Authors should carefully/accurately check the entire article to remove extensive typos and grammatical mistakes.
  • References list: The authors must follow the format depending on the MDPI-Electronics style. For instance, some papers' names in the references list begin with an uppercase letter in each word such as [1] and [2], and other references use an uppercase only in the first word such as [3] and 5]. The authors should remove the double quote from the reference name [85], some references do not contain enough information such as [67], [71], … etc. some journals names are not italic such as [85], [87], … etc. There are many references that have not accurate information, we think these references are unnecessary and should be removed. Some references are not strictly related to the topic of the research. Authors should fix all problems in the references list and scrutinize them carefully.

Author Response

Many thanks for the comments, critique, and suggestions. The response to all comments, critiques and suggestions are addressed and mentioned below:

  1. Overall Response: All concerns are appropriately addressed. A substantial revision is done. The size of the article is reduced. However, with the substantial improvements the important content is kept intact. I believe the new (revised- attached) manuscript is very compelling. The revised manuscript (attached) shows entirely different line numbers. Almost every section of the article content is revised, grammar checked, edited, and proofread. The revised manuscript strictly follows the MDPI-ASI template.

Examples:

  • An entirely new abstract is written (see line numbers 9 to 21 of the revised manuscript).
  • Since a new figure (figure number 2) is added in the methodology section, all the figure numbers are changed appropriately.
  • References/Citations are appropriately addressed.

  1. The abstract requires improvement, it is incomprehensive (The authors should briefly explain purpose/significance, method/process, and result/conclusion) and dependent (The abstract should be entirely understandable (self-contained text) on its own to a reader who has not read the whole article/associated research).

Response : A new comprehensive abstract (see line 9 to 21 in the revised manuscript) outlines the purpose, significance, methods, and conclusion bestowing the significant improvement of the previous abstract.

  1. We suggest that the authors should eliminate keywords such as “Internet of Virus Things” and “Theoretical Framework” because these keywords are already found in the article title. It is preferable that they replace them with other words to expand the reach of the search.

Response : (see lines 22/23 of the revised manuscript)-The keywords “Internet of Virus Things” and “Theoretical Framework” is deleted.” QBIoVT System Paradigm” is added. Quantum Vaccine Discovery is changed to QmRNA Vaccine Discovery.

  1. The introduction is comprehensive and contains a lot of valuable information, but it is a little long. It is preferable to reduce the size of the introduction section. Also, the authors' real names should be used instead of a word “author” (page2-line82, check whole article).

Response : ( see lines 25 to 72 of the revised manuscript)- The Introduction (section 1.) is shortened appropriately to make sure relevant and meaningful information kept intact. Line 54 (in the revised manuscript- old line number 82, page2) “a term coined by the authors” is deleted.   Throughout the article  “a term coined by the authors” is deleted.

  1. Section 1.2 (Significant Contribution) should improve, more detail is required because this section demonstrates to the reader the importance of this research.

Response : ( see lines 91 to 106 of the revised manuscript). Section 1.2 (Significant Contribution) is expanded to enhance the importance of this research.

  1. Some acronyms require definition before they are used, such as mRNA, IOE, FDA, EMA, ... etc.

Response: Corrected all over the article. Example: See lines 185-186 of the revised manuscript.

  1. The authors referred to a set of databases (page4-lines167-169), they claimed to be "double blind", in fact not all of them were "double blind", the authors should write the information accurately.

Response- (see line 129 of the revised manuscript) Corrected.

  1. In section 3.2.3 (IoVT/QmRNA Security, Privacy, Risk (page11)), the authors should provide some real-world examples of main security challenges.

Response - One example (authorization and authentication- lines 392 to 396 of the revised manuscript) is cited.

  1. The methodology requires some improvement.

Response: The Methodology is significantly improved. (See lines 433 to 511 of the revised manuscript). Content is added and a new figure added (figure 2) to describe the topology for which methodology is built.

  1. The authors noted the benefits of digitization (page 16). what about Searchability and availability.

Response:  Searchability and availability (Global Access Advantage) is added (see lines 616 to 620 of the revised manuscript).

  1. Section 9 requires some improvements. Although the authors pointed out the limitations of this study, they should also describe the findings of this study clearly and briefly.

Response: Section 9 ( Conclusion) is appropriately revised for improvements and to describe the findings.

  1. Figures and tables: The authors drew the figures and tables well. However, there are some issues. Figure 3 requires minimization. In Table 1.c and 1.d, some references do not contain a year such as [60] and [78]. In Table 2, the lines should be black color. In Figure 8, why are different colors of the arrow lines used? Figure 10 requires zooming in to be clearer.

Response:

Figure 3 is now figure 4, and correctly minimized.

Table 1c and 1d- [60] and [78] corrected.

Table 1c - [59] corrected. (Not suggested by the Reviewer) 

Table 2. Line colors corrected.

Figure 8. Line colors corrected. (In the revised manuscript Fig.9)

Figured 10. Zoom issue corrected. (In the revised manuscript Fig.11).

13.In general, this article is very long and requires reducing the size by removing unnecessary, duplicate sentences or improving writing.

Response: Size is reduced. Important content is kept intact. Writing is improved. Duplicate sentences are removed.

14.Proofreading: This article requires extensive proofreading to address English writing issues. There are many typos, grammar, and spelling issues that need extensive scrutiny. Also, some sentences, as well as paragraphs, are long that make the meaning unclear, the authors should address this issue. In particular, the authors should paraphrase many of the sentences (check the entire article) to make it distinct from the relevant papers and add necessary references if required. Authors should carefully/accurately check the entire article to remove extensive typos and grammatical mistakes.

Response:  The entire article is reviewed carefully. Almost every section of the article is revised. Proofreading is done as suggested. Typos, errors, grammar are corrected. Extensive editing is done. Typos, grammar, spelling issues are taken care of.

15.Reference’s list: The authors must follow the format depending on the MDPI-Electronics style. For instance, some papers' names in the references list begin with an uppercase letter in each word such as [1] and [2], and other references use an uppercase only in the first word such as [3] and 5]. The authors should remove the double quote from the reference name [85], some references do not contain enough information such as [67], [71], … etc. some journals names are not italic such as [85], [87], … etc. There are many references that have not accurate information, we think these references are unnecessary and should be removed. Some references are not strictly related to the topic of the research. Authors should fix all problems in the references list and scrutinize them carefully.

Response:  I believe all the issues of the Reference list is corrected and followed the MDPI format strictly.

Reference name [85] is corrected.

Reference [67], [71], is corrected.

References [85], [87], and others (throughout the Reference list) is corrected with regards to Italic issue.

Based on the extensive literature review, the reference list is appropriately corrected.

Reviewer 2 Report

The paper is very well written and an extensive review of the state of the art relating to sectors such as Quantum Theory, Quantum Mechanics, Internet of Virus Things, etc.

The authors focus on their theoretical description, highlighting how they are closely related to each other and include an extensive and recent bibliographic review. As also stated by the authors themselves, the article lacks, not only the empirical test of the proposed model (which would be fundamental to support the theory), but also the model itself is extremely generic and therefore not very concrete. For this reason, the work is too generic, it does not make real proposals and therefore it is not possible to evaluate the goodness of the model.

The authors claim to present an application of the IoVT for monitoring physical distance in pandemic situations. But no description, albeit superficial, of the system is provided. IoT nodes, smartphone applications and machine-learning algorithms for data analysis and diagnosis are listed, but the paper seems to be a merely a review of concepts and systems. Although the basic idea could also be interesting, given the topicality of the problem, the work lacks of additional details and indispensable insights into the real system architecture that the authors suggest to implement. Everything remains too vague.

Author Response

Many thanks for your comments, and critiques.

IMPORTANT NOTE: Most of the sections of the entire manuscript is revised to make it organic, tidy, and much more compelling.

A. The authors focus on their theoretical description, highlighting how they are closely related to each other and include an extensive and recent bibliographic review. As also stated by the authors themselves, the article lacks, not only the empirical test of the proposed model (which would be fundamental to support the theory), but also the model itself is extremely generic and therefore not very concrete. For this reason, the work is too generic, it does not make real proposals and therefore it is not possible to evaluate the goodness of the model.

Response:

With due respect, may I submit the following for your consideration:

1. Based on the significant insights on QBIoVT paradigm, foundational theories, theoretical and managerial significance as well as rationale outlined below confirm that the proposed novel theoretical framework is relevant, concrete and not generic.

a. An exhaustive review and synthesis of the QBIoVT elements (specifically mRNA, QI, IoVT) are presented.

b. The theoretical framework developed identifies QBIoVT challenges, priority areas, applications, and architecture empower researchers and practitioners alike to experience increased trust, benefits in the rapidly growing domain of the vaccine discovery, specifically in the unprecedented pandemic situation.

c. The framework guides the QBIoVT system initiatives.

d. The framework for QBIoVT adoption and execution are intended for researchers and practitioners.

e. The framework reveals opportunities for the QBIoVT applications for QmRNA vaccine discovery.

Thus, the theoretical framework developed identifies QBIoVT priority areas and challenges, providing a guide for those leading QmRNA initiatives and revealing opportunities for future research.

2. Even though the article lacks empirical test, the contribution with regards to theoretical and managerial significances, as outline in the sections of 1.2, 3, 7, are compelling. Furthermore, the proposed theoretical framework describes how pharma firms in vaccine discovery pursuit make their strategic decision by building the digitization building blocks and the topology (explained in the methodology section 4.2) analysis.

3. We believe, the framework developed will stimulate future research about QBIoVT system activities and will promote greater understanding of the future evolution of QmRNA vaccine discovery.

4. Research directions should be drawn from this proposed theoretical framework to propose new theories.

5. In our view, the theoretical framework developed in this paper falls under a theory’s focal phenomena that has not been covered by prior theories.

B. The authors claim to present an application of the IoVT for monitoring physical distance in pandemic situations. But no description, albeit superficial, of the system is provided. IoT nodes, smartphone applications and machine-learning algorithms for data analysis and diagnosis are listed, but the paper seems to be a merely a review of concepts and systems. Although the basic idea could also be interesting, given the topicality of the problem, the work lacks additional details and indispensable insights into the real system architecture that the authors suggest implementing.

Response:

Please see the sections: 3.2.1; 5.3; 5.3.1; 6.2; 6.2.1; and 6.2.2. New content is added (specifically see the lines 793 to 828 in the revised manuscript, specifically).

Round 2

Reviewer 1 Report

The authors have responded to most of our concerns, however, there are some comments that still require addressing by the authors.

  • The word “author” used in in two places (page7-line335 and page29-line291). In the first place there is no problem. In the second place, if the authors of this study are intended, then the authors should add "the authors of this study" to remove confusing.
  • Some acronyms require definition before they are used, such as IOE, ... etc.
  • Proofreading: This article still needs moderate proofreading to address English writing issues.
  • Paraphrasing: There are many sentences taken from previous research. Some of this research is included in the list of references, others are not (this is not acceptable). Authors must paraphrase all sentences taken from the previous research. The authors must avoid taking whole sentences from the original papers. They are obligated to paraphrase during the revision of the entire article. Also, authors should include missing references. For example,
    • “systems analysis and design, computer networking, information security, database management, and decision support systems.”
    • “When one particular gene needs to do its work, it makes a copy of itself, which is called messenger RNA,” and “mRNA is a fragile molecule, which means it must be coated in a protective, fatty covering to keep it stable and the refrigeration conditions has to do with how the mRNA was manufactured and stabilized.”
    • “DNA structures called plasmids, which transmit from one organism to the next in a process called conjugation”
    • “IoT empowers technological development and support to the vaccine distribution system. The impact of IoT will affect how mRNA vaccine supply chain leaders’ access information needed to improve the service. The IoT revolution can improve the supply chain”

.

.

.

Etc.

We indicated in the previous review that the authors must paraphrase some sentences in this paper. Once again, we recommend that the authors check the entire paper and paraphrase some of the sentences/paragraphs.

Author Response

Thanks for the comments, and critiques. The response to all comments, and critiques are addressed and mentioned below:

1. Overall Response: Manuscript is revised again based on the Reviewer1 comments and critiques. Further improvements are done. The new revised 2.0 dated March 21, 2021- manuscript is attached. The revised 2.0 manuscript dated March 22, 2021 shows again new line numbers. Almost every section of the article content is revised, grammar checked, edited, proofread, and strictly follows the MDPI-ASI template. Paraphrased throughout the article wherever required.

2. The word “author” used in in two places (page7-line335 and page29-line291). In the first place there is no problem. In the second place, if the authors of this study are intended, then the authors should add "the authors of this study" to remove confusing.

Response: Both Corrected. See sections 2.2.4 and 8.9

3. Some acronyms require definition before they are used, such as IOE

Response: Corrected. See Line # 64-65 (Section 1.0)

4. Proofreading: This article still needs moderate proofreading to address English writing issues.

Response: Further Proofreading done.

5. Paraphrasing: There are many sentences taken from previous research. Some of this research is included in the list of references, others are not (this is not acceptable). Authors must paraphrase all sentences taken from the previous research. The authors must avoid taking whole sentences from the original papers. They are obligated to paraphrase during the revision of the entire article.

Response: Paraphrasing done throughout the article. The references outlined in the revised manuscript is essential for the literature review. Based on the comments and critique of the review, I have changed few references appropriately. Thanks for your understanding.

6. Also, authors should include missing references. For example,

a. “systems analysis and design, computer networking, information security, database management, and decision support systems.”

Response: Corrected with appropriate citation and reference (Line # 155/156). Some Lines are Paraphrased.

b. “When one particular gene needs to do its work, it makes a copy of itself, which is called messenger RNA,” and “mRNA is a fragile molecule, which means it must be coated in a protective, fatty covering to keep it stable and the refrigeration conditions has to do with how the mRNA was manufactured and stabilized.”

Response: Corrected with appropriate citation and reference (Line 175 to 179)

c. “DNA structures called plasmids, which transmit from one organism to the next in a process called conjugation”

Response: Corrected with appropriate citation and reference (Line # 330-331)

d. “IoT empowers technological development and support to the vaccine distribution system. The impact of IoT will affect how mRNA vaccine supply chain leaders’ access information needed to improve the service. The IoT revolution can improve the supply chain”

Response: Corrected with appropriate citation and reference. (Line # 791 to 795)

Reviewer 2 Report

The authors have replied the comments, and I think the paper may be accepted in present form.

Author Response

Please see the revised manuscript dated March 21, 2021.

Thanks for your kind cooperation.